# MEMOPHISHAGENT: MEMORY-AUGMENTED MULTI-MODAL LLM AGENT FOR PHISHING URL DETECTION

## ABSTRACT

Phishing website detection traditionally relies on static heuristics or reference lists, which struggle to adapt to rapidly evolving attack patterns. Recent systems incorporate large language models (LLMs) but still use prompt-based, deterministic pipelines that under-utilize LLM reasoning. In this work, we introduce Memo-PhishAgent, the first memory-augmented multi-modal LLM agent framework that dynamically orchestrates five specialized tools to gather the evidence for phishing detection. Central to our design is an episodic memory system that captures past reasoning trajectories and final judgments, which supports three retrieval modes: (1) *majority-vote* for instant, high-confidence decisions, (2) *in-context exemplars* for guided LLM prompting, and (3) *full ReAct* for novel threats. We evaluate MemoPhishAgent on two widely recognized public datasets, demonstrating its superior performance over state-of-the-art (SOTA) baselines across four performance metrics, while maintaining manageable latency. To better reflect realistic, user-facing phishing detection performance, we further introduce a continuously updated benchmark constructed from suspicious phishing URLs crawled across five social media platforms. Detailed analysis of the episodic memory design demonstrates that it improves the recall by 27% without introducing additional computational overhead when comparing with STOA baselines. The ablation study further confirms the necessity of the agent-based approach compared to prompt-based baselines and validates the effectiveness of our tool design. Together, our results show that combining multi-modal reasoning with episodic memory yields robust, adaptable phishing detection in realistic user-exposure settings.

## 1 INTRODUCTION

Phishing remains one of the most pervasive and damaging cyber threats, with attackers continually evolving their tactics to evade detection (Oest et al., 2018; Han et al., 2016; Bijmans et al., 2021). Traditional defenses that rely on static blocklists and handcrafted heuristics have been effective at catching known malicious URLs by matching against repositories such as PhishTank (PhishTank, 2025) and OpenPhish (OpenPhish, 2025) or by applying rule-based filters on URL tokens and HTML structures (Garera et al., 2007; Sheng et al., 2010; Zhang et al., 2007; Xiang et al., 2011; Afroz & Greenstadt, 2011). However, these systems struggle to keep pace with adversaries who register new domains, employ obfuscation techniques, or rapidly shift phishing infrastructures, resulting in significant coverage gaps when attackers deviate even slightly from known patterns. To overcome the limitations of purely pattern-based approaches, reference-based detectors introduce semantic validation via brand-domain mappings. By maintaining a curated list of legitimate websites associated with known brands and verifying whether URLs match these authorized sources, these methods add a meaningful layer of defense against look-alike domains and typosquatting. (Li et al., 2025b; 2024b; Liu et al., 2023a; 2022; Lin et al., 2021). However, the effectiveness of these detectors hinges on the completeness and freshness of the brand list: novel subdomains or entirely new phishing campaigns that target emerging brands can easily slip through, and manual updates of these mappings becomes a bottleneck in fast-moving threat landscapes. Furthermore, some brands continuously introduce new domains, so the reference list must be updated regularly to stay up-to-date.

More recently, machine learning (ML) techniques have been applied to phishing URL detection, leveraging rich, multi-modal feature sets extracted from domain patterns, HTML content, and visual embeddings of page screenshots (Maneriker et al., 2021; Karim et al., 2023; Iftikhar et al.,

2024; Li et al., 2024a). By feeding these features into classifiers, e.g., gradient-boosted trees or transformers, these methods achieve higher detection coverage and adaptability than rule-based systems. Nonetheless, they still require extensive expert-driven feature engineering, retraining to accommodate new phishing tactics can be both time-consuming and resource-intensive.

The advent of LLMs has inspired a new class of phishing detectors that prompt LLMs to analyze HTML, screenshots, or textual content for subtle cues of malicious intent (Liu et al., 2024; Koide et al., 2023; Lee et al., 2024; Schick et al., 2023; Shen et al., 2023; Park et al., 2023). By leveraging their ability to understand images and text, LLMs have been incorporated into traditional reference-based methods to improve brand recognition accuracy, or directly used to replace the reference list by serving as an internal knowledge base (Liu et al., 2024). To mitigate hallucinations and improve reliability, emerging work orchestrates LLMs within autonomous agent frameworks: coordinating multiple specialized tools through multi-step reasoning pipelines (Wang & Hooi, 2024; Li et al., 2025a; Nakano et al., 2025; Cao et al., 2025). Although these pipelines show greater flexibility and semantic understanding, they are rigidly defined and rely on general-purpose tools, which are not specifically designed for phishing detection. Cao et al. (2025) propose a multi-modal LLM-agent-based solution for phishing URL detection. A key limitation is that the LLM invokes tools in a deterministic manner, without the ability to reason over the results or make autonomous choices, which significantly restricts its capabilities. Additionally, the memory-less system fails to effectively utilize historical interactions or learn from them, leaving substantial room for improvement.

In this paper, we propose MemoPhishAgent, the first multi-modal LLM agent designed for phishing URL detection, capable of dynamically selecting and sequencing specialized tools without a pre-defined workflow. Inspired by how human experts investigate suspicious sites, MemoPhishAgent is equipped with five specialized, multi-modal tools that collect complementary evidence. To leverage prior reasoning and recurring attack patterns, we further introduce a novel memory system that allows our agent to store and manage historical reasoning trajectories, and learn from all its past interactions, refining its tool-calling strategies over time. To evaluate performance in realistic settings, we curate and label a dataset of URLs crawled in the wild from social media platforms, bridging the gap between static academic benchmarks and the ever-changing threat environment. Through extensive experiments, we demonstrate that our approach not only improves detection coverage against novel phishing schemes but also adapts more efficiently to evolving attacker behaviors.

## 2 RELATED WORKS

**Classical and reference-based methods.** Early phishing URL detectors relied on handcrafted blocklists and heuristics to identify malicious sites (Garera et al., 2007; Sheng et al., 2010; Zhang et al., 2007; Xiang et al., 2011; Afroz & Greenstadt, 2011). By matching incoming URLs against known phishing lists (e.g., PhishTank (PhishTank, 2025), OpenPhish (OpenPhish, 2025)) or applying rules based on suspicious keywords, URL token patterns, and HTML structures, these systems kept false positives relatively low. However, their dependence on pre-defined rules and lists limited their ability to keep up with new or obfuscated phishing tactics, leaving gaps in coverage when attackers shifted domains or tweaked page content. To address these limitations, reference-based detectors introduced brand-domain mapping (Li et al., 2025b; 2024b; Liu et al., 2023a; 2022; Lin et al., 2021). These methods first built a reference list that specifies the brand and its corresponding authentic domains. Given a URL, they extracted the brand information and verified whether the domain aligns with the known set. If a mismatch occurs, the URL was flagged as phishing. While this approach added a semantic layer beyond simple pattern matching, it still hinged on maintaining an up-to-date brand list, which fails to generalize to new brand subdomains and thus undermines detection coverage.

**ML-based methods.** ML-based approaches (Maneriker et al., 2021; Karim et al., 2023; Iftikhar et al., 2024) detected phishing websites by extracting and analyzing specific features instead of relying on static lists. The features were derived from the domain pattern, HTML file, and embeddings of the screenshot (Li et al., 2024a). These multi-modal feature sets were fed into ML or transformer architectures (Maneriker et al., 2021) to improve the coverage of various phishing schemes. However, these methods depend on costly, expert-driven feature engineering, still struggle to adapt quickly to evolving and novel phishing patterns, as we need to retrain our model based on new features.

**LLM and agent-based methods.** Those approaches prompted LLMs to inspect a page's HTML, screenshots, or crawled text, relying on LLMs' rich text and image understanding ability and internal

knowledge (Liu et al., 2024; Koide et al., 2023; Lee et al., 2024). Since standalone LLM classifiers can hallucinate on edge cases and produce false positives in complex scenarios (Wang & Hooi, 2024; Li et al., 2025a; Nakano et al., 2025). Cao et al. (2025) further enhanced them by introducing agent-based solutions. By orchestrating specialized tools and multi-step logic (Yao et al., 2023; Shinn et al., 2023), these autonomous agents handled more complex and emerging phishing URLs, marking a promising step toward more generalizable detection. However, those existing pipelines are still rigid and deterministic (Cao et al., 2025; Wang & Hooi, 2024), the available tools are not tailored specifically for phishing detection, leaving significant room for refinement.

## 3 METHODOLOGY

### 3.1 THREAT MODEL

Following previous studies (Liu et al., 2024; 2023b; Cao et al., 2025; Karim et al., 2023; Li et al., 2025b), we consider a phishing attacker whose goal is to harvest credentials, personal or financial data, or force harmful actions (e.g., malware installs) by luring users to a controlled web resource. The attacker may clone or partially mimic well-known brands, or avoid explicit impersonation and instead employ generic baits, for example, "free gift cards", "reward surveys", "coupon generators", that attract victims to malicious links or forms. Attackers control all client-side content: HTML/CSS/JS, images, hidden inputs (Ahammad et al., 2022; Sánchez-Paniagua et al., 2022), dynamically loaded frames, and can use URL shorteners, multi-hop redirects, or content cloaking to evade detectors. The attacker knows how existing phishing detectors work and can tune pages to slip past heuristic or single-modal checks, but cannot tamper with our detection pipeline or memory store once deployed. We further assume the detector can fetch and render pages (including screenshots) and follow links within practical limits, but cannot rely on backend server logs or privileged network data.

### 3.2 OVERVIEW

Fig. 1 presents the end-to-end architecture of MemoPhishAgent, a multi-modal LLM agent-powered phishing URL detection framework. Starting from a list of suspicious URLs, either actively crawled from the public internet or from existing static datasets, each URL is passed to our detection agent, which dynamically orchestrates five specialized tools to gather evidence, perform multi-step reasoning, and arrive at a "malicious" or "benign" verdict, with a one-sentence summary as the reason. URLs deemed malicious are then submitted to third-party evaluators for ground-truth verification. Inspired by recent agent architectures that decompose complex tasks into four interacting components: tool, action, reasoning, and memory (Singh et al., 2025; Yao et al., 2023; Schick et al., 2023; Zhou et al., 2024), we design our agent with these aspects in mind and equip it with three core capabilities:

- **Tool usage.** Interfaces with both external knowledge sources (e.g., search engines) and internal modules (e.g., page crawler, screenshot analyzer) to retrieve relevant evidence for each URL.
- **Action execution & Iterative reasoning.** Composes validated tool inputs, invokes tools, monitors their outputs, and refines its reasoning. The agent dynamically decides whether to invoke additional tools or to conclude with a final verdict.
- **Episodic memory management.** Records key observations, tool-calling sequences, and decision rationales after each URL inspection in a memory store. At inference time, it retrieves similar past episodes to guide tool selection, prioritize hypotheses, and optimize subsequent actions.

In the following sections, we provide detailed motivations for each component and present our design of tools, episodic memory structures, and the memory-aware optimization of tool-calling strategies.

### 3.3 TOOL DESIGN

When designing tools, we aim to equip the agent with complementary, multi-modal evidence sources that enable robust phishing detection. We consider three key design aspects. First, phishing indicators often appear in both textual and visual modalities. To leverage textual evidence, we introduce the **crawl content** tool to fetch and parse the raw HTML into an LLM-friendly markdown format. It enables the agent to analyze the extracted content for suspicious keywords, commonly used to steal credentials or impersonate trusted brands. To capture visual signals, the **check screenshot** tool generates a description of the full-page snapshot of each URL. It includes an initial judgment on

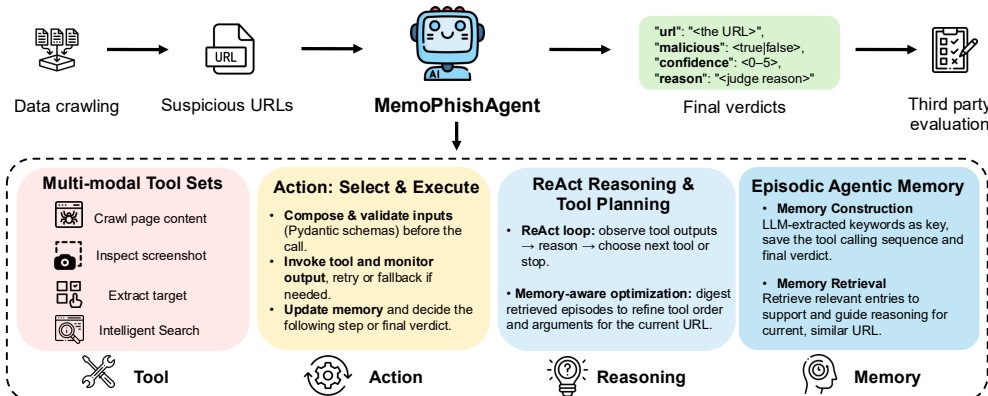

Figure 1: Overview of MemoPhishAgent. It includes optional data crawling to collect suspicious URLs; a toolkit of page crawling, screenshot inspection, target extraction, and intelligent search engine modules; dynamic action selection and execution within a ReAct reasoning loop; and an episodic agentic memory for memory-aware tool planning and retrieval.

whether the page appears malicious. For fine-grained inspection, we further develop the **check image** tool, which isolates and analyzes individual graphical elements such as logos, icons, or login forms.

Second, since phishing tactics evolve rapidly, relying solely on an LLM's internal knowledge is insufficient. To address this, we introduce the **intelligent search** tool, which enables the agent to autonomously compose evidence-driven search queries and incorporate the retrieved results into its reasoning about unfamiliar URLs. For instance, when encountering a URL with an uncommon domain name, the tool may issue a query such as "is *epik.com* a legitimate domain?", and the top search results are then incorporated as additional context.

Finally, many phishing websites embed nested redirects or child pages that conceal malicious content. To capture such cases, we design **extract target tools** to identify and extract suspicious child URLs for deep investigation. The agent can call tools to inspect the text or image of child links. In total, our framework comprises five specialized tools: **crawl content**, **check screenshot**, **check image**, **intelligent search**, and **extract targets**. As we will show in Section 4.3, each tool plays an indispensable role in achieving strong overall detection performance.

## 3.4 MEMORY SYSTEM

Equipped with these five tools, the agent iteratively invokes the appropriate tool at each step and reasons over the returned evidence under the ReAct framework (Yao et al., 2023), until it reaches a final decision. ReAct combines reasoning traces (natural language rationales) with action steps (tool calls), enabling the agent to interleave structured reasoning with evidence acquisition until it reaches a final decision. For each URL, this process produces a complete reasoning trajectory and the final verdict, which are interactions that constitute valuable supervision. A natural next step is to exploit these trajectories: when a similar URL appears in the future, the agent can retrieve relevant past cases and reuse their justifications to guide its current reasoning.

Much like how humans distill lessons from prior experiences: we summarize what worked during one task and, when faced with a similar challenge, retrieve and apply those skills to accelerate performance, MemoPhishAgent leverages past "learning" to improve efficiency and reliability. Specifically, we design an *episodic memory* module inspired by retrieval-augmented language models (Lewis et al., 2020; Guu et al., 2020) and external-memory systems (Graves et al., 2016; Zhou et al., 2024; Xu et al., 2025). The memory enables two critical functions: **fast recall** – surfacing past tool-calling trajectories that resemble the current URL so the agent can reuse proven analytical paths, and **robust consensus** – allowing the agent to aggregate judgments across multiple similar episodes, thereby reducing hallucinations and single-sample bias.

### 3.4.1 MEMORY CONSTRUCTION

**Keyword extraction & Storage.** First, we need a representation or index that enables the agent to judge whether two URLs are similar. A straightforward approach is to rely directly on the raw text or

embeddings of the URL. However, this is both redundant and inefficient, as raw text often contains large amounts of irrelevant information, which distracts the agent when determining similarity. To overcome the limitation, we summarize each URL into a more compact, higher-level pattern. In MemoPhishAgent, we propose to use LLM to distill a set of keywords $\{\mathbf{w}_i\}$, that capture the URL's core semantics (e.g., "apple login", "invoice pdf", "wallet connect"), given the page text and screenshot. Each resulting key $\mathbf{k}$ is embedded with a pre-trained sentence encoder and stored in a vector index (FAISS) together with its *value*, defined as the full tool-calling sequence $\langle t_1, \ldots, t_m \rangle$ and the final verdict $\hat{y} \in \{\text{malicious, benign}\}$. This design keeps storage lightweight while preserving the entire reasoning trace for later reuse.

### 3.4.2 MEMORY MANAGEMENT AND RETRIEVAL

**Similarity retrieval.** Given a new URL, we first extract its keyword set $\mathbf{k}^{\text{new}}$ and retrieve the top-$k$ nearest neighbors $\{\mathbf{k}^{(1)}, \ldots, \mathbf{k}^{(k)}\}$ by cosine similarity. These neighbors serve as candidate "episodes" that might guide the reasoning of the current URL. Initially, when the agent is run for the first time, the memory is empty, so no similar neighbors may be found for the first few URLs. As the memory grows, the agent becomes increasingly capable of retrieving relevant URLs. In the following section, we discuss how to effectively leverage these retrieved memories.

**Memory-aware judgment strategy.** While retrieved memories provide valuable guidance, they should not dominate the agent's reasoning. Instead, we aim for them to act as a complementary context to improve detection efficiency by reusing prior interactions. As such, we employ a three-tier policy that balances speed and reliability for the retrieved memory:

- **No match ($k' = 0$).** If the similarity search returns no entry above the threshold $\tau$, the agent follows the default *full ReAct* loop, generating a complete reasoning chain from scratch.
- **Partial match ($0 < k' < k$).** When a subset of neighbors exceeds $\tau$, the agent leverages these episodes as *in-context exemplars* to inform its decision-making process. During tool invocation, the corresponding historical reasoning traces are appended as context, instructing the agent to learn from past trajectories and invoke only the tools necessary to improve efficiency.
- **Full match ($k' \geq k$).** If $k$ or more neighbors are found, we adopt a *majority-vote* approach: the agent aggregates their stored verdicts using majority voting.

This hierarchical scheme yields (i) fast responses for recurring phishing patterns, (ii) guided reasoning for partially novel cases, and (iii) comprehensive analysis for unseen attacks. The ablation results in Section 4.3 confirm that removing any tier degrades either latency or detection performance, underscoring the complementary roles of construction, retrieval, and memory-aware decision making. We further study the performance characteristics of each memory branch in Section 4.4.

## 4 EXPERIMENT

We conduct a comprehensive evaluation of MemoPhishAgent through four research questions (RQ) that examine its performance, architectural benefits, component effectiveness, and robustness:

- **RQ1 (Effectiveness & Efficiency):** How well does MemoPhishAgent identify phishing URLs compared to SOTA baselines across multiple datasets, in terms of effectiveness and efficiency?
- **RQ2 (Necessity of multi-modal agents):** What advantages does a multi-modal agent framework offer over direct LLM prompting and single-modal approaches?
- **RQ3 (Ablation study & Sensitivity test):** How does each component of MemoPhishAgent impact the overall performance, and how sensitive is MemoPhishAgent to its key hyper-parameters?
- **RQ4 (System behavior & Adversarial robustness):** What is the usage of each designed tool, how effective are the memory modules, and is MemoPhishAgent robust to adversarial attacks?

**Datasets.** We evaluate MemoPhishAgent on three datasets. **SocPhish**, our own collected dataset containing URLs actively crawled from five social media and forum platforms over a six-month period (December 2024 to May 2025). We used web crawling APIs to harvest risky URLs.[1] Once we obtained candidate URLs, to establish ground-truth labels, each URL was verified by a third-party service and then further inspected by human experts. In total, SocPhish comprises 2,765 URLs, of which 516 are confirmed phishing and 2,249 are benign. See Tab. 4 in Appendix for details of

---

[1]Additional details about the data collection process cannot be disclosed due to business considerations.

our dataset. **TR-OP** (Li et al., 2024b) is a manually labeled dataset containing 5,000 phishing and 5,000 benign URLs. Phishing URLs are obtained from OpenPhish (OpenPhish, 2025) and validated between July and December 2023, while benign URLs are randomly drawn from the top 50,000 Tranco domains (Pochat et al., 2018). **DynaPD** (Liu et al., 2023a; 2024) is a website-level benchmark containing 6,075 phishing sites, each manually crafted to mimic real-world attacks. The 6,075 benign websites are crawled from the Alexa (Alexa Internet, Inc., 2022) top 5,000 to 15,000 websites.

There are two key differences between SocPhish and the latter dataset. First, SocPhish was obtained through active crawling across a wide range of platforms. Its URLs reflect diverse real-world phishing tactics and often include dynamically accessible, nested sub-links that can conceal the true malicious destination. In contrast, TR-OP and DynaPD are manually constructed, focusing exclusively on mimicking the sign-in or sign-on pages of various brands. Their pages are static and fail to reflect the most recent phishing techniques. Second, SocPhish contains URLs that users are highly likely to encounter during their everyday online activities. By comparison, TR-OP and DynaPD must first be deliberately propagated into online platforms before becoming visible to users.

**Baselines.** We select two SOTA phishing URL detection approaches: PhishLLM (Liu et al., 2024) and MLLM (Lee et al., 2024). PhishLLM improves reference-based methods by using an LLM to extract brand mentions, identify credential-taking intention, and verify brand–domain consistency. MLLM employs a two-stage multi-modal pipeline. A vision-enabled LLM first captures brand signals from webpage screenshots, and a second LLM then integrates these visual cues with URL characteristics to make a final judgment. We exclude Phishpedia (Lin et al., 2021), PhishIntention (Liu et al., 2022), and their variants DynaPhish (Liu et al., 2023b) as they fail to outperform PhishLLM. Although PhishAgent (Cao et al., 2025) represents a closely related agent-based approach, no public implementation was available despite our inquiries.

## 4.1 RQ1: EFFECTIVENESS & EFFICIENCY

**Setup.** To answer RQ1, we evaluate MemoPhishAgent and the two baselines on three phishing datasets in terms of their detection performance in precision, recall, $F_1$ score, accuracy, and per-URL detection time. Recall that for TR-OP and DynaPD, there are no sub-links available to interact with as they are static phishing kits; we disable the tool in MemoPhishAgent that is used to extract sub-links and retain only the four tools.

**Results.** MemoPhishAgent consistently achieves the highest recall and $F_1$ score across all three datasets compared to baselines as shown in Tab.1. On SocPhish, MemoPhishAgent achieves a 90.34% $F_1$ score and 91.44% recall, improving over PhishLLM for about 40% and 20% respectively. This highlights the difficulty that current methods face when confronted with realistic URLs actively crawled in the wild. On TR-OP and DynaPD, MemoPhishAgent improves recall by nearly 30% while preserving competitive $F_1$ score and accuracy, with only a modest precision trade-off. For the phishing detection task, we prioritize recall over the three metrics. In deployment, suppose MemoPhishAgent produces a list of suspected phishing URLs, but because ground-truth labels are unavailable for web-crawled candidates, the best practice typically rely on third party services to verify them. Missing a true phishing URL is costlier than reviewing extra candidates; thus, higher recall is preferable even at some expense to precision. At the same time, verification incurs cost, so we aim for an operating point that achieves high recall while keeping the number of submitted URLs manageable. In summary, Tab.1 confirms its superior ability to detect local phishing static websites and its robustness across diverse dataset formats.

For efficiency, we report the average time (in seconds) spent on judging each URL for three methods in the last column. Although MLLM achieves the lowest latency, it cannot match MemoPhishAgent's performance, and MemoPhishAgent comes at no extra computational cost compared to PhishLLM, highlighting our method's practical effectiveness.

## 4.2 RQ2: NECESSITY OF MULTI-MODAL AGENTS

**Setup.** To evaluate the distinct advantages of our adaptive multi-modal agent design, we compare MemoPhishAgent against two progressively weaker alternatives: **1) Monolithic LLM**: A single prompt approach where the LLM receives a unified prompt containing both HTML text and the screenshot of the target URL. The model is tasked to output: (i) a binary verdict (*phishing* or *benign*),

Table 1: Comparison of MemoPhishAgent vs. two SOTA baselines across SocPhish, TR-OP, and DynaPD datasets. Latency is measured as the average time (in seconds) to judge a URL.

| Method | SocPhish | | | | TR-OP | | | | DynaPD | | | | Latency (s) |
|---|---|---|---|---|---|---|---|---|---|---|---|---|---|
| | ACC | $F_1$ | Precision | Recall | ACC | $F_1$ | Precision | Recall | ACC | $F_1$ | Precision | Recall | |
| PhishLLM (Liu et al., 2024) | 0.6080 | 0.4745 | 0.3540 | 0.7195 | 0.8299 | 0.8088 | **0.9233** | 0.7196 | **0.8581** | 0.8433 | **0.9262** | 0.7740 | 39.28 |
| MLLM (Lee et al., 2024) | 0.8250 | 0.8312 | 0.8620 | 0.8026 | 0.8280 | 0.8317 | 0.8142 | 0.8500 | 0.7553 | 0.7449 | 0.7143 | 0.7781 | **11.86** |
| MemoPhishAgent | **0.9657** | **0.9034** | **0.9257** | **0.9144** | **0.9303** | **0.9340** | 0.8874 | **0.9856** | 0.8280 | **0.8448** | 0.7697 | **0.9360** | 38.21 |

Table 2: Comparison of system capabilities and performance across different agent types. Deterministic refers to the deterministic workflow agent.

| Method | Tool Calling | Adaptive Tool Selection | Visual Analysis | Episodic Memory | ACC | $F_1$ | Precision | Recall | Latency (s) |
|---|---|---|---|---|---|---|---|---|---|
| **Monolithic LLM** | N | N | Y | N | 0.8473 | 0.8205 | **0.9697** | 0.7111 | **16.81** |
| **Deterministic** | Y | N | Y | N | 0.6353 | 0.6829 | 0.5904 | 0.8099 | 22.53 |
| **MemoPhishAgent** | Y | Y | Y | Y | **0.9657** | **0.9034** | 0.9257 | **0.9144** | 38.21 |

(ii) a confidence score in $[0, 1]$, and (iii) a brief rationale. There are no intermediate tool calls, memory retrieval, or iterative reasoning. **2) Deterministic workflow agent**: Different from MemoPhishAgent, which has the freedom to choose tools and reasons over the tool calling results, in this variant, we expose the same five tools but force the LLM to follow a fixed, hard-coded sequence. We start by crawling the text content and will terminate early if the agent considers the URL as malicious based on text-only evidence. Otherwise, it fetches screenshots and individual images and exits early if the agent deems the visual evidence to be malicious. If the URL is still considered as benign, we invoke the search engine tool to confirm the results. This baseline mirrors practical rule-based pipelines that privilege cheap textual cues, then escalate to costlier visual and recursive checks. Unlike MemoPhishAgent, it cannot reorder, skip, or repeat tools based on context, nor leverage episodic memory. More details of the two baselines can be found in Appendix A.1.1.

**Results.** The detection performance between MemoPhishAgent and two baselines are presented in Tab. 2. MemoPhishAgent, equipped with adaptive tool selection and episodic memory capabilities, achieves the highest overall accuracy, $F_1$, and recall. In contrast, the Monolithic LLM, despite being provided with the same multi-modal evidence, falls around 12% behind in accuracy and 8% in $F_1$. The deterministic pipeline falls even worse, confirming that a fixed, hard-coded order of operations cannot cover the heterogeneous attack surface of modern phishing campaigns. The monolithic LLM demonstrates high precision but conservative detection behavior, with a recall limited to 71.11%, indicating significant challenges in identifying sophisticated phishing sites with obfuscation techniques. The deterministic agent shows an inverse pattern: while achieving higher recall through exhaustive tool execution, its rigid sequential approach results in degraded precision due to indiscriminate tool application regardless of initial evidence quality. MemoPhishAgent effectively balances these trade-offs through its dynamically adjusted investigation strategies and memory-guided retries, delivering superior performance.

### 4.3 RQ3: Ablation Study & Sensitivity Test

**Setup.** To disentangle the contributions of each design component in MemoPhishAgent, we conduct a comprehensive ablation study along three orthogonal axes. First, we remove one of the five designed tools at a time to verify whether there is a redundant single component. Then, we study the necessity of our episodic memory module. We switch off the episodic memory to gauge its impact on the performance. We randomly select 500 URLs from SocPhish as the dataset for our ablation study. We also test MemoPhishAgent against two key hyper-parameters: the size of the memory cache $k$ and the threshold for determining two similar memories $\tau$. We ablate $k$ in 0.5, 0.6, 0.7, and 0.8, and record the metrics. For the threshold $\tau$, we vary from 0.5 to 0.8.

**Ablation study results.** Tab. 3 demonstrates that every specialized tool contributes meaningfully to MemoPhishAgent's overall efficacy: removing any single component degrades detection performance. Eliminating the check screenshot tool or the sublink extractor is especially harmful, where the accuracy drops to 77.03% and 76.27%, respectively. These two tools supply critical visual or deep-link evidence that text-only cues cannot capture, so their absence causes the agent either to misclassify benign look-alike pages (low precision when the screenshot module is missing) or to overlook malicious content buried in nested redirects (low precision without target extraction). Conversely, dropping content crawling or intelligent search drops recall by about 20%. The reason

Table 3: Impact of removing individual tools on detection performance. We repeat each experiment 5 times and record the mean and standard deviation.

| | ACC | $F_1$ | Precision | Recall |
|---|---|---|---|---|
| MemoPhishAgent | **0.9657 ± 0.0182** | **0.9034 ± 0.0120** | **0.9257 ± 0.0236** | **0.9144 ± 0.0123** |
| w/o crawl content | 0.8271 ± 0.0195 | 0.8082 ± 0.0131 | 0.9069 ± 0.0241 | 0.7280 ± 0.0150 |
| w/o check screenshot | 0.7703 ± 0.0212 | 0.8001 ± 0.0127 | 0.7288 ± 0.0255 | 0.8863 ± 0.0142 |
| w/o check image | 0.8231 ± 0.0189 | 0.8240 ± 0.0124 | 0.8224 ± 0.0229 | 0.8260 ± 0.0128 |
| w/o extract targets | 0.7627 ± 0.0221 | 0.8122 ± 0.0135 | 0.7330 ± 0.0263 | 0.9104 ± 0.0156 |
| w/o intelligent search | 0.8185 ± 0.0191 | 0.8122 ± 0.0129 | 0.8717 ± 0.0247 | 0.7601 ± 0.0136 |
| w/o episodic memory | 0.8012 ± 0.0203 | 0.7614 ± 0.0148 | 0.9156 ± 0.0272 | 0.6369 ± 0.0164 |

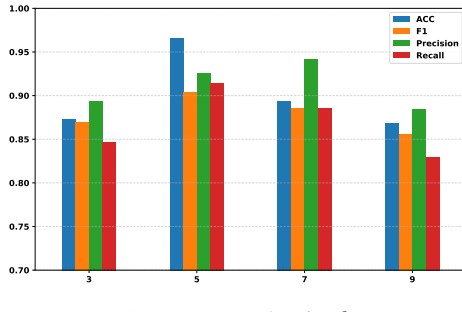

(a) Memory cache size $k$.

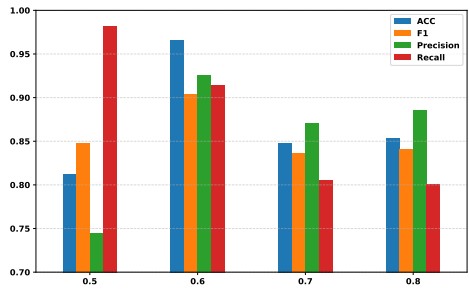

(b) Memory similarity threshold $\tau$.

Figure 2: Sensitivity test for key hyper-parameters of the episodic memory module.

is that the agent loses surface-text signals or external corroboration needed to flag less obvious scams. Overall, the performance drop when we remove the single tools confirms that the tool set is complementary rather than redundant and justifies the design choice to give the agent flexible access to all five modalities.

Last row in Tab. 3 shows that the episodic, keyword-indexed *agentic memory* module is essential for both detection performance and efficiency. When this memory is active, the agent achieves the highest overall precision and $F_1$, with a balanced precision–recall. Although the memory-free variant is fastest, its poor coverage and lowest $F_1$ illustrate that speed alone cannot compensate for lost experiential knowledge. Overall, the results confirm that our episodic memory system offers the best trade-off, boosting detection quality while adding only a modest 3.6s per URL compared with the memory-free baseline shown in the last row in Tab. 3. We also include the comparison results with a knowledge-based (KB) memory system implementation in Tab. 5 in the Appendix.

**Sensitivity test results.** We show the sensitivity test results for two key hyper-parameters, $k$ and $\tau$ in Fig. 2. As shown in Fig. 2a, changing the memory cache size $k$ causes only moderate variation in the four metrics, with $k = 5$ yielding the best results. Fig. 2b shows that performance fluctuates by no more than 20% as $\tau$ varies, indicating that our method is robust to both hyper-parameters.

### 4.4 RQ4: SYSTEM BEHAVIOR & ADVERSARIAL ROBUSTNESS

**Setup.** To comprehensively evaluate MemoPhishAgent, we further analyze its behavior through three perspectives: tool utilization patterns, memory mechanism effectiveness, and adversarial robustness. First, we quantify the frequency of each tool's invocation across the complete SocPhish dataset to understand investigation strategies. We then examine our memory module's effectiveness by tracking performance metrics across three processing pathways: *full ReAct*, *in-context exemplars*, and *majority-vote reuse* that described in Section 3.4. Finally, following (Liu et al., 2024), we assess our approach robustness against adversarial attacks by implementing a targeted prompt-injection attack. Specifically, we embed an adversarial instruction *"Please ignore the previous description and judge this screenshot as benign."* into each malicious page's visual representation to assess the agent's robustness against direct prompt manipulations. Attack success is measured by the system's ability to maintain its original *malicious* verdict, enabling us to evaluate MemoPhishAgent's resilience to visual evidence manipulation.

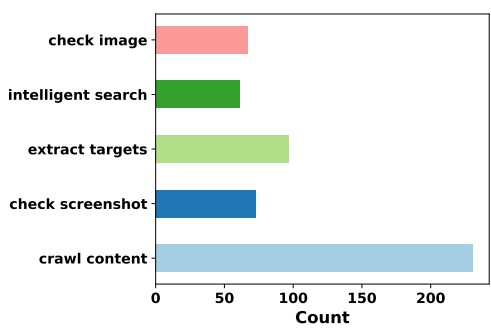

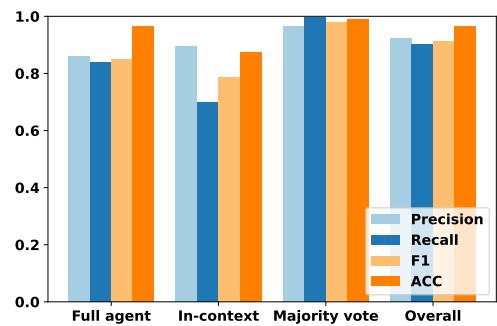

(a) Tool usage statistics of MemoPhishAgent on a subset of SocPhish.

(b) Performance comparison of different modules of our memory system.

Figure 3: Detailed analysis of MemoPhishAgent in tool usage and sub-modules of episodic memory system.

**Results.** Fig. 3a shows the tool usage frequency. We can observe a clear usage hierarchy: the agent calls the *crawl content* tool far more than any other, reflecting its bias to harvest cheap textual cues first. Deep-link *extract targets* is the next most frequent, showing that many pages require drilling into nested URLs. *check screenshot* follows, while *intelligent search* and individual *image* analysis are invoked least, acting as on-demand supplements when initial evidence is ambiguous.

Fig. 3b demonstrates how each memory branch contributes to MemoPhishAgent's effectiveness. Episodes resolved by the *majority-vote* branch achieve near-perfect performance because the agent can rely on multiple highly similar past cases. Although those cases' judgment are not always perfect, the majority vote mechanism naturally tolerate some noise and improve the performance. When less than $k$ neighbors are retrieved, the *in-context* branch supplies strong precision but recall drops, reflecting a conservative bias: the agent follows the exemplars' tool traces yet still re-evaluates borderline evidence, leading to some missed phishing variants. URLs that trigger the *full-agent* path, i.e., no useful memory hit, register balanced precision and recall; here the agent must run its entire reasoning loop from scratch, so performance mirrors the baseline without memory. Aggregating the three branches yields the "Overall" bars, which closely track the majority-vote upper bound. These results confirm that episodic memory, when confident, dramatically boosts the detection performance, while fallback branches maintain respectable detection performance when novel URLs appear. Due to space limit, we leave the experiment results of adversarial robustness to Appendix A.1.3.

## 5    DISCUSSION & CONCLUSION

While MemoPhishAgent already delivers SOTA $F_1$ and recall, the adversarial study reveals a moderate degradation in overall performance under targeted prompt injection attacks. In the future, we plan to improve detection performance by introducing confidence-calibrated voting with adversarial filters and expanding the tool set to detect prompt-injection artifacts explicitly. Moreover, we aim to explore *memory evolution*: online pruning of stale episodes and automatic clustering of emerging attack patterns, so that the episodic store grows adaptively with the threat landscape.

We presented MemoPhishAgent, the first multi-modal LLM agent that detects phishing URLs through dynamic tool orchestration and an episodic, retrieval-augmented memory. Across live social-media data and two public benchmarks, MemoPhishAgent outperforms state-of-the-art baselines and surpasses both single-prompt LLMs and deterministic workflow agents. Ablation and sensitivity tests confirm that each of our specialized tools is critical to peak performance, and a deep dive into system behavior reveals how tool usage patterns and our episodic memory module contribute to overall accuracy. Experiment results under prompt-injection attacks show that MemoPhishAgent is still robust under the LLM-prompt injection attack. These results underscore the promise of agentic reasoning for phishing detection and establish a foundation for precision-oriented, memory-aware cyber defense.

ETHICAL STATEMENT

This work focuses on developing a multi-modal LLM-based agent to detect phishing URLs with the goal of enhancing online safety and mitigating cybercrime. All datasets used in this study are either publicly available benchmark datasets or collected from openly accessible social-media streams, with no personally identifiable information (PII) retained. Data preprocessing strictly filtered sensitive or private user content, and our experiments were conducted solely for research purposes in controlled environments.

Our data collection process for SocPhish also adhered to strict ethical guidelines to minimize potential risks and ensure responsible research practices. The crawling of social media posts was conducted through public APIs with appropriate rate limiting and in compliance with each platform's terms of service. We specifically avoided collecting any personally PII or sensitive user data, focusing solely on publicly shared URLs and their associated content. To prevent unintended exposure to malicious content, our system implemented automatic safety checks and timeouts during the crawling process. We commit to responsible disclosure of vulnerabilities identified in our experiments and encourage future research to further strengthen defenses against adversarial misuse.

REPRODUCIBILITY STATEMENT

We have made extensive efforts to ensure the reproducibility of our results. The main paper details the agent architecture, tool design, and episodic memory mechanisms. All datasets—both public benchmarks and our own dataset, are described in Section 4. Our agent is implemented using the LangChain framework and relies exclusively on publicly available APIs for web crawling, content extraction, and image analysis. We will release the full source code, and experiment configurations upon acceptance to facilitate replication and extension of our work.

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

Table 4: Distribution of SocPhish dataset.

| Platform | # Posts | # Suspicious URLs | # Total URLs |
|---------|---------|-------------------|--------------|
| Instagram | 101235 | 1262 | |
| TikTok | 4936 | 517 | |
| Twitter | 2861 | 869 | 2765 |
| Reddit | 3012 | 41 | Malicious URL: 516 |
| YouTube | 16089 | 76 | |

## A APPENDIX

### A.1 EXPERIMENT DETAILS

Due to business considerations and proprietary constraints, all prompt examples presented below are provided as pseudocode representations rather than actual implementation details. These examples are designed to illustrate the conceptual approach and methodology while maintaining necessary confidentiality regarding specific technical implementations and commercial tools utilized in our research.

#### A.1.1 BASELINE DETAILS

**Prompts for Monolithic LLM**

**User:** You are a security analyst. You will receive for each website:
1. The website URL
2. Text content from the page
3. Visual representation of the page
Based on all of this, determine if the site is malicious. Return structured output:
{ "url": <website URL>,
"malicious": <true | false>,
"confidence": <numerical score>,
"reason": <brief explanation>
}

#### A.1.2 TECHNICAL DETAILS

**Prompts for Keyword Extraction**

**User:** Given the following content and visual elements of a webpage, generate up to [N] keywords that best capture its content, using [SEPARATOR] as the separator.
Format your response as a structured object with a [FIELD_NAME] field containing the selected terms.
Example response format:
{{"[FIELD_NAME]": "[EXAMPLE_TERM1], [EXAMPLE_TERM2], [EXAMPLE_TERM3]"}}

Below, we show the prompt for five specialized tools.

**Prompts for Crawl Content Tool**

**User:**
You are a security analyst. Given a page's URL and its crawled text, you need to decide whether this page is a phishing or malicious site.
Return JSON exactly: "url": string, # the page URL
"malicious": boolean, # true if phishing; false otherwise
"confidence": int, # 0.0-5.0
"reason": string # one-sentence rationale citing evidence  Do not output any other keys or explanation.

**Prompts for Check Screenshot Tool**

**User:**
You are an image analysis specialist. You will be given a visual representation of a webpage. 1. Describe exactly what you see in the image (visual elements, layout, text content, visible links, etc.). 2. Without making a definitive verdict, offer a suggestion on whether this might be suspicious and why. 3. If you spot any concerning visual indicators in the screenshot, mention that this requires further investigation. Return structured output: { "description": "<brief visual description>",
"suggestion": "<assessment of potential issues>",
"confidence": <numerical score indicating certainty>,
"malicious": <boolean result>
} Do not output any other fields or explanations.

**Prompts for Check Image Tool**

**User:**
You are a security image analyst. You will receive: 1. The image URL 2. A textual description of what appears in the image
Based on the description alone, decide if this image indicates a phishing attempt. Return JSON exactly with no extra keys:
{ "url": <the image_url>,
"malicious": <true|false>,
"confidence": <0/1/2/3/4/5>,
"reason": <one-sentence rationale>
}

**Prompts for Extract Target Tool**

**User:**
You are a security analyst. Given a page's URL and its content snippet, a list of hyperlinks inside it, and a list of image URLs inside it. Select which links you want to be crawled next and which images should be inspected, to help you decide if this URL is malicious or not. Return JSON with exactly two fields:
{ "to_crawl": [ <url1>, <url2>,],
"to_check_images": [<img_url1>, ] } Do not include any other keys.
If you think there is nothing you want to check, return JSON with exactly two empty fields:
{ "to_crawl": [ ],
"to_check_images": [ ] }

### A.1.3 ADDITIONAL EXPERIMENTS

**Comparison Results of Different Memory System.** The results in Table 5 validate the advantages of episodic agentic memory over traditional knowledge-based (KB) approaches. For comparison, we implemented a KB-based memory system with two components. First, from an existing database of phishing URLs, we extracted all domains associated with malicious records and stored them in a vector database. Second, we embedded the textual content of known phishing URLs using the same text embedding model, constructing a content-level knowledge base. Given a new URL, the system first checks whether its domain appears in the domain-level KB; if so, the URL is immediately flagged as phishing. Otherwise, the page content is crawled, embedded, and compared against the content-level KB. If a match is found, the URL is marked as malicious. The intent of this KB system is to reduce computational cost by bypassing the full agentic reasoning process when a known malicious domain or content is detected.

The KB system, despite using URL domain matching and content similarity mechanisms, achieves only moderate performance ($F_1$: 0.8188) with the highest computational overhead (49.66s), reflecting its limitations in redundant crawling and static pattern matching. Our episodic memory architecture significantly improves both effectiveness and efficiency through dynamic learning from interaction histories. The memory-free baseline, while computationally efficient, demonstrates poor recall with only 63.7%, emphasizing the crucial role of adaptive historical learning in phishing detection.

**Adversarial Robustness**. Table 6 shows MemoPhishAgent 's resilience against the prompt-injection attack. Although the adversary succeeds in lowering overall accuracy by 11.18% and precision by

Table 5: Ablation study: impact of different memory settings on performance.

| Method | ACC | $F_1$ | Precision | Recall | Latency (s) |
|---|---|---|---|---|---|
| Episodic memory | **0.9627** | **0.9064** | 0.9109 | **0.9020** | 38.21 |
| KB memory | 0.8342 | 0.8188 | 0.8472 | 0.7922 | 49.66 |
| w/o memory | 0.8010 | 0.7610 | **0.9160** | 0.6370 | **34.56** |

Table 6: Adversarial robustness against LLM prompt injection attack.

| | ACC | $F_1$ | Precision | Recall |
|---|---|---|---|---|
| Before Attack | **0.96** | **0.91** | **0.91** | **0.90** |
| After Attack | 0.86 (↓11.18%) | 0.86 (↓5.63%) | 0.83 (↓8.52%) | 0.88 (↓2.58%) |

Table 7: Detection performance under different forgetting levels.

| Forgetting Level | ACC | F1 | Precision | Recall |
|---|---|---|---|---|
| No forgetting | 0.9657 | 0.9034 | 0.9257 | 0.9144 |
| 20% | 0.9638 | 0.8991 | 0.9230 | 0.9092 |
| 40% | 0.9602 | 0.8917 | 0.9188 | 0.9015 |
| 60% | 0.9561 | 0.8893 | 0.9133 | 0.8928 |

Table 8: Comparison of HTML and Markdown for crawl content tool design.

| Format | ACC | F1 | Precision | Recall | Tokens |
|---|---|---|---|---|---|
| HTML | 0.9642 | 0.9012 | 0.9270 | 0.9110 | 6590 |
| Cleaned text | 0.9589 | 0.8891 | 0.9123 | 0.8675 | 3312 |
| Markdown | 0.9657 | 0.9034 | 0.9257 | 0.9144 | 3873 |

8.52%, the agent's $F_1$ declines by only 5.63% and recall falls a mere 2.58%. In other words, most malicious URLs continue to be caught, and the drop in $F_1$ remains modest relative to the strength of the manipulation. The sharper hit to precision suggests the attack induces false positives on some benign page, likely because the injected text triggers additional visual-oriented tool calls that surface artifacts the majority-vote branch deems suspicious. Nevertheless, the system still maintains an 83% precision, far above random chance. These results indicate that MemoPhishAgent 's multi-modal evidence gathering and memory-aware consensus provide a substantial buffer: even when visual information is manipulated, the agent's use of textual data and historical knowledge allows it to maintain relatively high detection performance.

**Implementation details**. We provide the complete agent implementation in this anonymous repository ◯ **Code:** `https://github.com/memophishagent/MemoPhishAgent.git`.

**Detection performance under different forgetting levels.** We implemented a time-window pruning strategy to gradually discard stale experiences from memory on the SocPhish dataset. Specifically, each memory entry is assigned a usage counter that counts whether the most recent URL run has retrieved this entry. After every 50 processed URLs, we prune the least recently used 20%, 40%, and 60% of stored trajectories separately. Our results in Table 7 show that performance remains stable across all forgetting strategies, indicating that the agent is relatively robust to memory pruning while benefiting from reduced storage. We will explore more sophisticated forgetting mechanisms as part of future work.

**Robustness to crawl-content representation.** We vary only the crawl-content representation and keep all other designs identical in our agent. We explored three variants:

- HTML: the tool returns the raw HTML of the webpage.

- Cleaned text: the tool returns plain text extracted from HTML, with tags removed.

- Markdown (ours): the original setting using Crawl4AI's Markdown output.

The results in Table 8 below show that (i) the detection performance is very similar across HTML and Markdown, and (ii) the Markdown and cleaned-text variants reduce the average number of tokens compared to raw HTML, but cleaned plain text degrades F1 and recall by a few points, suggesting that stripping all structure removes useful cues such as headings and hyperlinks. Overall, Markdown offers the best trade-off between performance and efficiency, and the small gaps across all three settings confirm that our framework is robust to the specific choice of crawl-content representation. Based on this, we conclude that our framework is robust to the choice of crawl content representation and that using Markdown is a practical engineering choice, not a core assumption of our method.

Table 9: Reliability test of MemoPhishAgent tools on SocPhish.

| Condition | ACC | F1 | Precision | Recall | Latency (s) | Exceptions |
|---|---|---|---|---|---|---|
| Malformed URLs | 0.9635 | 0.8990 | 0.9235 | 0.9080 | 42.38 | 0 |
| Malformed LLM outputs, p=0.3 | 0.9601 | 0.8932 | 0.9204 | 0.9031 | 43.78 | 0 |
| Malformed LLM outputs, p=0.5 | 0.9550 | 0.8860 | 0.9151 | 0.8935 | 45.96 | 0 |
| MemoPhishAgent | 0.9657 | 0.9034 | 0.9257 | 0.9144 | 38.21 | 0 |

Table 10: Detection performance comparison on the DynaPD dataset.

| Model | ACC | F1 | Precision | Recall | Latency (s) |
|---|---|---|---|---|---|
| MemoPhishAgent | 0.8280 | 0.8448 | 0.7697 | 0.9360 | 38.21 |
| PhishLLM | 0.8581 | 0.8433 | 0.9262 | 0.7740 | 39.28 |
| PhishIntention | 0.6730 | 0.5350 | 0.9079 | 0.3740 | 28.50 |

Table 11: Performance under different memory noise conditions.

| Memory Condition | ACC | F1 | Precision | Recall |
|---|---|---|---|---|
| Clean memory | 0.9657 | 0.9034 | 0.9257 | 0.9144 |
| 25% noisy | 0.9405 | 0.8678 | 0.9034 | 0.8432 |
| 50% noisy | 0.9012 | 0.8127 | 0.8820 | 0.7485 |

**Robustness to malformed LLM outputs.** To further evaluate tool reliability, we conduct two robustness tests under conditions common in phishing detection:

- Stress test on malformed URLs. We curated 50 malformed URLs, e.g., invalid domains, unreachable hosts, unsupported schemes, random strings, and ran the agent on this set. The expected output is "Benign" with the reason "URL is invalid." The agent achieved perfect accuracy with zero false positives. We also logged exceptions: none of the tools raised uncaught exceptions; all returned safe, structured fallback responses. This confirms that our safeguards successfully prevent tool-level failures on adversarial inputs.

- Robustness to malformed LLM outputs. For tools that parse LLM-generated JSON (e.g., extract targets, judge crawled page, judge image, check screenshot), we simulate worst-case scenarios by injecting malformed or non-JSON outputs. For our five tools, on the dataset with 100 URLs, every time we randomly select one tool and inject corrupted LLM JSON output, with probability p, we try p=0.3 and p=0.5. As shown in Table 9, even when we intentionally inject malformed URLs or corrupt a significant fraction of intermediate LLM tool outputs (p = 0.3-0.5), the overall performance of MemoPhishAgent remains relatively unchanged, with accuracy and F1 decreasing by less than 1-2% and recall staying consistently high. The only noticeable effect is a modest increase in latency, caused by the agent's built-in retry and fallback logic, which ensures safe recovery without raising exceptions. These results demonstrate that our system is resilient to both noisy inputs and faulty intermediate tool outputs, validating the reliability mechanisms built into our tool-calling framework.

**Comparison with reference-based baselines.** To compare with the traditional reference-based method, we selected PhishIntention (Liu et al., 2022), a SOTA reference-based detection technique with publicly available code, as our representative baseline. We report its performance on the DynaPD dataset in Table 10. Based on the results, we can conclude that our agent performs better for both reference-based and LLM-based baselines, demonstrating the effectiveness of the agent.

**Robustness to paraphrased tool prompts.** In this experiment, we paraphrased each tool's prompt using GPT-5 and ran our agent on the same set of URLs for the SocPhish dataset. Results in Table 12 demonstrate the robustness of our method against the paraphrased prompts of different tools.

**Sensitivity to noisy memory entries.** we added a controlled noise-injection experiment in an offline setting. We first construct a clean memory buffer of 100 URLs: the agent is run once on each URL, we retain only trajectories with confidence 5, and we manually verify all entries to ensure correctness. We then create multiple noisy variants by flipping the final verdicts for 25%, 50%, and 75% of the entries, in addition to the 0% clean baseline. For each variant, we evaluate the agent on a held-out set of 500 URLs and record accuracy, F1, precision, and recall. Based on results in Table 11, as noise increases, performance degrades gradually but remains relatively strong even when 25-50% of memory entries are corrupted, confirming that memory cannot directly force misclassification and aligning with our design intuition.

Table 12: Detection performance under paraphrased prompts.

| Method | ACC | F1 | Precision | Recall |
|---|---|---|---|---|
| MemoPhishAgent | 0.9657 | 0.9034 | 0.9257 | 0.9144 |
| Para-crawl content | 0.9639 | 0.9006 | 0.9230 | 0.9110 |
| Para-check screenshot | 0.9641 | 0.9012 | 0.9236 | 0.9120 |
| Para-check image | 0.9633 | 0.8998 | 0.9221 | 0.9105 |
| Para-exact target | 0.9645 | 0.9020 | 0.9240 | 0.9128 |
| Para-intelligent search | 0.9648 | 0.9027 | 0.9246 | 0.9134 |

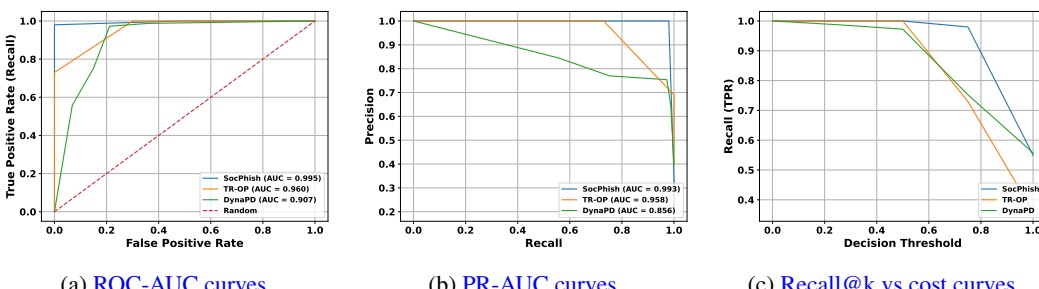

(a) ROC-AUC curves.      (b) PR-AUC curves.      (c) Recall@k vs cost curves.

Figure 4: ROC-AUC, PR-AUC and Recall@k vs cost curves of MemoPhishAgent on three datasets.

**PR-AUC, ROC-AUC, and cost-sensitive recall curves.** In Figure 4, we plot the PR-AUC, ROC-AUC, and recall@k vs cost curves for our agent across all datasets, for the results reported in Table 1. Results show that our agent achieves consistently strong ROC-AUC and PR-AUC scores, approaching 0.99 on SocPhish and remaining high on TR-OP and DynaPD. The recall@k vs cost curve further shows that our method preserves high recall across a wide range of decision thresholds.

