# OpenReview forum: "MemoPhishAgent: Memory-Augmented Multi-Modal LLM Agent for Phishing URL Detection"
_ICLR.cc/2026/Conference — Submitted to ICLR 2026_

### Official Review · Reviewer_39jv · 2025-10-26

**Soundness:** 3
**Presentation:** 3
**Contribution:** 3
**Rating:** 6
**Confidence:** 3

**Summary:**

This paper introduces MemoPhishAgent, a novel memory-augmented, multi-modal LLM agent framework designed for phishing URL detection. The agent dynamically orchestrates five specialized tools to gather evidence. Its core contribution is an episodic memory system that stores and retrieves past reasoning trajectories to inform current decisions. This memory supports three retrieval modes: majority-vote for recurring threats, in-context exemplars for similar cases, and a full ReAct loop for novel threats. To validate their approach, the authors introduce a new dataset, SocPhish, crawled from social media. Experiments conducted on SocPhish and two public benchmarks demonstrate that MemoPhishAgent outperforms state-of-the-art baselines, particularly in recall.

**Strengths:**

- The core idea of an agent that learns from its own reasoning history (episodic memory) to improve phishing detection is novel and promising.
- The paper provides a thorough evaluation on three different datasets, including a newly created one that reflects a more realistic threat landscape. The ablation studies effectively demonstrate the value of each component.
- The method shows substantial improvements over SOTA baselines, especially in recall, which is a critical metric for this security application.

**Weaknesses:**

- The reported average latency of \~38 seconds per URL poses a major challenge for practical, real-time deployment. This is significantly slower than the MLLM baseline (\~12s) and would likely be too slow for scanning large volumes of URLs in a production environment. The paper does not offer a clear path to mitigating this critical bottleneck.
- The paper overlooks the crucial aspect of memory management and scalability. In a real-world scenario, the episodic memory would grow indefinitely, leading to increased retrieval times and storage costs. The lack of a strategy for pruning, summarizing, or managing this growing memory store is a significant omission in the proposed framework.

**Questions:**

- The proposed method is over 3 times slower than the MLLM baseline. Could you provide a more detailed analysis to justify this significant increase in latency? In what specific real-world scenarios would this trade-off be acceptable? Are there plans to reduce this latency to a more practical level?
- What is your long-term vision for managing the episodic memory? Without a mechanism for pruning or summarizing, the system's performance will inevitably degrade over time. Can you propose a concrete strategy to address this?

---

> ### Author Response · Authors · 2025-11-24
> **Author Responses - Part 1**
>
> We thank the reviewer for their insightful suggestions and questions. We have responded to their individual points below.
>
> **The reviewer questioned that the latency of the proposed method would be challenging for practical deployment, when compared to the MLLM baseline, and asked for practical plans to reduce the latency.**
>
> We sincerely appreciate the reviewer’s question regarding the latency–performance trade-off.
> The MLLM baseline achieves an 11.86s latency through a fixed two-stage pipeline: a vision-enabled LLM analyzes the screenshot, then another LLM integrates visual cues with URL features for the final judgment. This workflow only includes two steps, thus it does not introduce too much latency, but it is also strictly deterministic and cannot gather additional evidence when signals are incomplete or ambiguous.
>
> In contrast, our agent performs adaptive, multi-step reasoning, dynamically invoking tools such as crawl content, check screenshot, visual analysis, extract targets, and intelligent search. When initial evidence is insufficient or contradictory, the agent can proactively gather complementary information, for example, conducting fine-grained logo or form analysis or extracting nested URLs to inspect obfuscated redirect chains. Those additional and necessary tools calling and output analysis, increased the latency of our method, in order to achieve the superior detection performance (11.18 percentage points higher recall (91.44% vs. 80.26%)). while the MLLM baseline lacks this capability.
>
> Regarding the 38s latency, we clarify that in our experiment, we process each URL one by one in a sequential manner, which was constrained by our LLM API rate limits. In practice, our agent is designed to support parallel execution: with sufficient API quota, multiple worker instances can process URLs concurrently, allowing throughput to scale linearly with available resources. In a production deployment with adequate parallelism, the effective per-URL latency would be significantly reduced. We will add those analyses in the updated version.
>
> We outlined several practical plans for reducing per-URL latency without changing the core architecture:
> - Parallel and asynchronous tool execution. As discussed above, now the latency is calculated for sequential processing or URLs, in the future, we will explore parallelization, which can reduce per-URL latency by up to $2-3×$ in real deployments.
> - Cached rendering and incremental crawling. We can store the cached rendering, especially for previously appeared benign pages, to reduce render time for the crawl content tool. For the specific crawl content tool that we select, enabling cache can save about 2 seconds for one URL. We can also enable the domain caching to avoid re-evaluating the URL within a preset time window, which could further reduce latency.
>
> Taking all of the practical engineering optimizations into consideration, we can see that the latency can be substantially improved in a production setting.
>
>
> **The reviewer asked for a concrete, long-term plan for managing the episodic memory, e.g., summarization or pruning.**
>
> We sincerely appreciate the reviewer's thoughtful question. We will first outline our design choices that naturally constrain memory size, then summarize the long-term plan.
>
> First, our system design naturally provides strong robustness against the memory poisoning attack. First, only evaluations with maximum confidence (score = 5) enter episodic memory, establishing a quality threshold that makes low-confidence poisoning attempts ineffective. Second, the three-tier retrieval strategy employs majority-vote aggregation when multiple similar cases exist ($k \geq 5$), providing natural robustness against outliers; a single poisoned entry cannot override consensus from legitimate historical cases. Third, the similarity threshold ($\tau$) prevents loosely related malicious entries from influencing unrelated evaluations. Lastly, in order to make the memory poisoning effective, the attacker needs to manipulate multiple entries, which further degrades the stealthiness of the attack, making detection far more likely.
>
> To provide concrete evidence of storage efficiency: at our production scale of 5,000 URLs daily with 80% memory retention, the system stores about 4,000 entries per day, or 360,000 entries over a 90-day window. Assuming an average of ~1 KB per entry (URL plus compact metadata), that’s roughly 360 MB of raw data. Even scaling to 100,000 daily URLs with the same 80% retention would yield ~7.2 million entries over 90 days, or about 7.2 GB of raw data at 1 KB per entry. Even after accounting for database and index overhead, this remains well within practical limits for modern infrastructure.

---

> > ### Author Response · Authors · 2025-11-24
> > **Author Responses - Part 2**
> >
> > Regarding the long-term production deployment for memory management, we propose the following three concrete strategies:
> > - Least Recently Used (LRU) Eviction Policy: We will implement an LRU-based memory management system that tracks access patterns and automatically removes the least recently retrieved episodes when storage limits are approached. This approach is particularly well-suited for phishing detection because attack campaigns exhibit short lifecycles, since recent patterns remain highly relevant while older tactics become obsolete.
> >
> > Implementation Details: (1) Access tracking: Maintain timestamps for each episode's last retrieval during similarity search. (2) Capacity thresholds: Set soft limits (e.g., 5M episodes or 5GB storage) that trigger LRU-based pruning when reached, removing the bottom 20% of least-accessed entries. (3) Time-based bounds: Combine LRU with temporal constraints (e.g., automatically remove episodes older than 90 days regardless of access frequency) to ensure stale attack patterns don't persist indefinitely. (4) Graceful degradation: When episodes are evicted, the system seamlessly falls back to full ReAct reasoning without memory guidance, ensuring continuous operation without performance cliffs. This strategy maintains bounded memory while preserving detection effectiveness, and the frequently-accessed patterns for current threats remain available, while obsolete signatures that no longer match incoming URLs are systematically removed based on actual usage patterns rather than arbitrary time windows alone.
> >
> > - Preliminary of LRU pruning. To further address the reviewer’s concern about removing outdated information, we implemented a preliminary time-window pruning strategy to gradually discard stale experiences from memory on the SocPhish dataset. Each memory entry is assigned a usage counter, and after every 50 processed URLs, we prune the least recently used 20%, 40%, and 60% of stored trajectories separately. Our results in Table 1 below show that performance remains stable across all forgetting strategies, indicating that the agent is robust to memory pruning while benefiting from reduced storage. We will explore more sophisticated forgetting mechanisms as part of future work.
> >
> > **Table 1. Detection performance under different forgetting levels.**
> > | Forgetting Level |   ACC   |   F1    | Precision | Recall |
> > |------------------|---------|---------|-----------|--------|
> > | No forgetting    | 0.9657  | 0.9034  | 0.9257    | 0.9144 |
> > | 20%              | 0.9638  | 0.8991  | 0.9230    | 0.9092 |
> > | 40%              | 0.9602  | 0.8917  | 0.9188    | 0.9015 |
> > | 60%              | 0.9561  | 0.8893  | 0.9133    | 0.8928 |
> >
> >
> > - Memory summarization. Instead of saving the raw tool calling sequences and each tool’s outputs, we add an extra step and ask the LLM to summarize the key takeaways for each entry, to improve memory efficiency.
> >
> > We will include all of the above discussions in the revised PDF.

---

### Official Review · Reviewer_QobJ · 2025-10-27

**Soundness:** 3
**Presentation:** 3
**Contribution:** 2
**Rating:** 4
**Confidence:** 4

**Summary:**

The paper introduces MemoPhishAgent, a memory-augmented multimodal LLM agent for phishing detection. It dynamically orchestrates five tools — text crawling, screenshot analysis, image inspection, target-link extraction, and intelligent search — within a ReAct loop. The agent employs an episodic memory module based on keyword retrieval and a three-tier reasoning strategy (majority voting → in-context examples → full ReAct). Evaluations on both public benchmarks (TR-OP, DynaPD) and a proprietary SocPhish dataset demonstrate improved F1 and recall performance over prior state-of-the-art methods.

**Strengths:**

### 1. Well-designed tool suite for the task.
The proposed five-tool composition — text crawling, screenshot inspection, image verification, target-link extraction, and intelligent search — is well aligned with the phishing-detection problem. Each tool is justified and its contribution is evaluated through ablation experiments (Table 3).

### 2. Clearly layered memory strategy.
The three-tier policy (majority voting → in-context exemplars → full ReAct reasoning), combined with keyword-based summarization and a FAISS index for retrieval, is conceptually clear and technically sound.

### 3. Real-world data.
The use of the SocPhish dataset, collected from social-platform contexts, shows a commendable effort to reflect realistic “user-exposure” scenarios beyond standard public benchmarks (TR-OP, DynaPD).

I believe that as LLMs and MLLMs continue to advance, this task becomes increasingly important and challenging. The authors’ data collection effort and their proposed new approach show promising potential.

**Weaknesses:**

### 1. Lack of the novelty
The paper repeatedly claims to be the first memory-augmented multi-modal LLM agent, yet prior work on memory-enabled or multi-tool agents (also cited by the authors) already explored similar structures. The distinction from previous “deterministic” or “static” agents is described conceptually, not experimentally. No quantitative comparison to these baselines under identical conditions is provided.

### 2. Limited and Potentially Biased Baselines
Only two LLM-based baselines (PhishLLM, MLLM) are compared. Strong traditional or hybrid methods (e.g., URLTran) are excluded on the basis of “inferior performance” or “unavailable code,” which undermines fairness.

“Reference-based” knowledge systems are said to underperform and thus are omitted, even though MemoPhishAgent’s majority-voting memory is conceptually similar to such retrieval paradigms.

### 3. Metric Reporting Bias
Evaluation focuses on a single operating point optimized for recall/F1. For a balanced understanding, PR-AUC, ROC-AUC, and cost-sensitive recall curves should be reported.

### 4. Inconsistency of LLM Agent Behavior
A notable issue—unaddressed in the paper—is the inconsistency of the LLM agent’s reasoning and action selection across identical or semantically similar inputs.
Because the ReAct loop depends on stochastic LLM generations without deterministic control (e.g., temperature > 0), the same sample may produce different tool sequences and conflicting conclusions.
This inconsistency undermines reliability, especially in a security-critical application such as phishing detection.

The authors should: Quantify decision variance under repeated inference (e.g., 10 runs per sample).
Evaluate robustness under prompt paraphrasing and equivalent representations.
Clarify if any sampling control or output normalization (e.g., function-call schema enforcement) was applied.

Without such measures, MemoPhishAgent cannot guarantee stable judgments, which is a key weakness in deployment contexts.

**I suggest additional experiments for strong paper.**

1. Broaden baselines (URLTran, hybrid URL + LLM models, recent reference-based retrieval methods).
2. Memory hygiene tests with controlled noise injection and forgetting strategies.
3. Report PR-AUC, ROC-AUC, Recall@k vs cost curves.

**Questions:**

Minor Comments
Typo “PhishGuardAgent” (Section 4.4). (I think this is MemoPhishAgent)

---

> ### Author Response · Authors · 2025-11-24
> **Author Responses - Part 1**
>
> We thank the reviewer for their insightful suggestions and questions. We have responded to their individual points below.
>
> **The reviewer raised the concern about the limited novelty of the paper compared to existing work, then noted that the paper offers only conceptual distinctions from earlier deterministic or static agents and provides no quantitative comparisons under identical conditions.**
>
> We appreciate the reviewer’s comments and suggestions. Our work differs from existing works fundamentally in tool structure and agent design. First, we have the memory module to enable the agent to learn effectively from past experiences. We also conduct ablation studies on both the episodic memory and an offline, knowledge-based memory variant, demonstrating their contribution to performance. Prior work does not include mechanisms for storing or leveraging past learning experiences.
>
> We would like to clarify that our distinctions between MemoPhishAgent and the “deterministic/static” baselines are not only conceptual but also supported by concrete definitions and quantitative evaluations already included in the paper. The deterministic workflow agent is formally defined in Lines 321-339, and the monolithic LLM baseline is defined in Lines 339-347, with their tool availability and execution constraints explicitly specified. To ensure a fair comparison, we design the same tools for both baselines. Furthermore, Section 4.1 - 4.3 presents a complete quantitative comparison under identical experimental conditions, evaluating accuracy, precision, recall, F1 score, and per-URL detection time across all three datasets. These results directly measure the performance gaps between MemoPhishAgent and both baseline architectures.
>
> **The reviewer raised the concern that the baseline selection is limited and suggested comparing with reference-based systems. They also pointed out that the majority-voting mechanism in memory is conceptually similar to the reference-based baselines.**
>
> We thank the reviewer for pointing out the related works. Regarding URLTran, it requires about 1M URLs to train the model according to the third-party implementation, and there are no public model checkpoints; as a result, reproducing the model fairly is not feasible. Instead, we selected PhishIntention[1], a reference-based method with publicly available code, as a representative baseline. We report its performance on the DynaPD dataset in Table 1 below. Based on the results, we can conclude that our agent performs better for both reference-based and LLM-based baselines, demonstrating the effectiveness of the agent.
>
> **Table 1. Detection performance comparison on the DynaPD dataset.**
>
> | Model            |   ACC   |   F1    | Precision | Recall | Latency (s) |
> |------------------|---------|---------|-----------|--------|---------|
> | MemoPhishAgent   | 0.8280  | 0.8448  | 0.7697    | 0.9360 | 38.21   |
> | PhishLLM         | 0.8581  | 0.8433  | 0.9262    | 0.7740 | 39.28   |
> | PhishIntention   | 0.6730  | 0.5350  | 0.9079    | 0.3740 | 28.50   |
>
> We would also like to clarify that majority voting and reference-based methods are fundamentally different mechanisms:
> - What is retrieved? In our framework, majority voting operates within the memory module: the retrieved items are past reasoning trajectories, along with two additional retrieved cases used as contextual memory. In contrast, reference-based methods retrieve predefined reference pages or templates that the target URL is compared against.
> - Purpose and decision logic. First, the memory in MemoPhishAgent is auxiliary: it provides supporting context but does not determine the final label. The agent still bases every decision on real-time tool outputs. For reference-based methods, they rely entirely on the similarity between the target URL and stored reference items; classification depends directly on these matches.
>
>
> [1] Liu et al., Inferring Phishing Intention via Webpage Appearance and Dynamics: A Deep Vision Based Approach, USENIX 2022.

---

> ### Author Response · Authors · 2025-11-24
> **Author Responses - Part 2**
>
> **The reviewer suggested adding PR-AUC, ROC-AUC, and cost-sensitive recall curves.**
>
> We thank the reviewer for the suggestions. Our agent outputs a binary predicted label (phishing or benign) and a discrete confidence in {1,2,3,4,5}, where the confidence score reflects the confidence in the predicted label, not the probability of phishing. For example, a prediction (label=0, confidence=5) means “very confident benign” and a prediction (label=1, confidence=5) means “very confident phishing”.
>
> To construct ROC-AUC, PR-AUC, and cost-sensitive recall curves, we convert this paired output $(\hat{y}, c)$ into a unified phishing-likelihood score. Specifically, we first normalize confidence via $c’=(c-1)/4$ to obtain values in $[0,1]$. We then map confident phishing predictions to high scores and confident benign predictions to low scores using $s=c’$, where $\hat{y} = 1; s=1-c’$, where $\hat{y} = 0$, so higher $s$ indicates higher phishing risk. This transformation preserves the model’s confidence structure while producing a monotonic phishing score suitable for threshold-sweeping metrics. Using this standardized score, we compute PR-AUC, ROC-AUC, and cost-sensitive recall curves across all datasets, for the results reported in Table 1 in the paper. For the other two baselines, in their detection methods, they do not design the method to generate any confidence scores; thus, those three plots are unavailable.
> The results are included in [**this link**](https://drive.google.com/drive/folders/1hDfvW39pSIeY0hgbb8BMxd1PjGNuWM7Y?usp=drive_link).
> We also added the analysis to our revised PDF. Results show that across all datasets, our agent achieves consistently strong ROC-AUC and PR-AUC scores, approaching 0.99 on SocPhish and remaining high on TR-OP and DynaPD. It demonstrates that the agent maintains excellent separability between phishing and benign URLs even under distribution shifts. The recall@k vs cost curve further shows that our method preserves high recall across a wide range of decision thresholds, which aligns well with our practical goals. These results highlight the robustness of our agent’s multi-tool reasoning pipeline and its ability to remain recall-dominant across diverse and realistic threat environments.

---

> > ### Author Response · Authors · 2025-11-24
> > **Author Responses - Part 3**
> >
> > **The reviewer raised the concern about the stochastic nature of ReAct generation. They argued that this instability is unaddressed for security-critical phishing detection and suggested quantifying variance across repeated runs, testing robustness to paraphrased inputs, and clarifying any sampling controls or schema constraints.**
> >
> > We thank the reviewer for raising this important point regarding consistency in LLM-based agents. First, we clarify that our reported results already aggregate 5 independent runs for every method and every dataset (as stated in Section 4.3). We report the mean and standard deviation, and these deviations are consistently small across metrics, indicating that the MemoPhishAgent exhibits low decision variance in practice, despite the inherent stochasticity of LLMs.
> >
> > Second, MemoPhishAgent reduces variability by design through several mechanisms already built into the system:
> > Schema-constrained tool calls (via LangChain function-calling) substantially restrict output space and eliminate free-form randomness. Deterministic tool behaviors (crawler, screenshot extractor, image parser, vector search) ensure that any non-LLM components behave identically across runs.
> > Structured system prompts and a strictly bounded tool selection space (five tools) sharply limit degrees of freedom in the ReAct loop. The final classification must reflect evidence synthesized from tool outputs, not from unconstrained generative reasoning, further reducing variance. These design choices mean that, although the agent uses ReAct-style reasoning, the potential for divergent trajectories is substantially narrower than unconstrained LLM reasoning pipelines.
> >
> > To address the reviewer’s concern, we included an additional experiment to test the robustness under paraphrased and semantically equivalent prompts. We paraphrased each tool’s prompt using GPT-5 and ran our agent on the same set of URLs for the SocPhish dataset. Results in Table 2 demonstrate the robustness of our method against the paraphrased prompts of different tools.
> >
> > **Table 2. Detection performance under paraphrased prompts.**
> > | Method                   |   ACC   |    F1    | Precision | Recall |
> > |--------------------------|---------|----------|-----------|--------|
> > | Original                 | 0.9657  | 0.9034   | 0.9257    | 0.9144 |
> > | Para-crawl content       | 0.9639  | 0.9006   | 0.9230    | 0.9110 |
> > | Para-check screenshot    | 0.9641  | 0.9012   | 0.9236    | 0.9120 |
> > | Para-check image         | 0.9633  | 0.8998   | 0.9221    | 0.9105 |
> > | Para-exact target        | 0.9645  | 0.9020   | 0.9240    | 0.9128 |
> > | Para-intelligent search  | 0.9648  | 0.9027   | 0.9246    | 0.9134 |
> >
> > Lastly, regarding the sampling controls, our tool-calling operates under a temperature of 0, and the reasoning text uses a temperature of 0.1; we will include all of those implementation details in the updated version.

---

> > > ### Author Response · Authors · 2025-11-24
> > > **Author Responses - Part 4**
> > >
> > > **The reviewer asked about the robustness of the memory system to noise and suggested performing hygiene tests with controlled noise and adding experiments about the forgetting mechanism.**
> > >
> > > We thank the reviewer for the question. We first clarify that in our design, episodic memory stores only high-confidence trajectories and is used solely through in-context learning and majority voting. Memory entries cannot directly cause misclassification: every final decision must still be supported by current tool outputs, and memory is never treated as ground truth. This design already provides an implicit form of memory hygiene suitable for online deployment.
> > >
> > > To make this explicit, we added a controlled noise-injection experiment in an offline setting. We first construct a clean memory buffer of 100 URLs: the agent is run once on each URL, we retain only trajectories with confidence 5 (as in the paper), and we manually verify all entries to ensure correctness. We then create multiple noisy variants by flipping the final verdicts for 25%, 50%, and 75% of the entries, in addition to the 0% clean baseline. For each variant, we evaluate the agent on a held-out set of 500 URLs and record accuracy, F1, precision, and recall. Based on results in Table 3, as noise increases, performance degrades gradually but remains relatively strong even when 25-50% of memory entries are corrupted, confirming that memory cannot directly force misclassification and aligning with our design intuition.
> > >
> > > **Table 3. Performance under different memory noise conditions.**
> > > | Memory Condition |   ACC   |   F1    | Precision | Recall |
> > > |------------------|---------|---------|-----------|--------|
> > > | Clean memory     | 0.9657  | 0.9034  | 0.9257    | 0.9144 |
> > > | 25% noisy        | 0.9405  | 0.8678  | 0.9034    | 0.8432 |
> > > | 50% noisy        | 0.9012  | 0.8127  | 0.8820    | 0.7485 |
> > >
> > > To explore memory forgetting, we designed a time-window pruning to gradually remove the outdated experiences of memory, under our current setup, where we will remove the least recently used 20%, 40% and 60% of the trajectories. Our preliminary results performance remains stable for forgetting strategies, demonstrating that our agent is robust to memory forgetting. We will leave the design of a more complex forgetting mechanism as our future work.
> > >
> > > **Table 4. Detection performance under different forgetting levels.**
> > > | Forgetting Level |   ACC   |   F1    | Precision | Recall |
> > > |------------------|---------|---------|-----------|--------|
> > > | No forgetting    | 0.9657  | 0.9034  | 0.9257    | 0.9144 |
> > > | 20%              | 0.9638  | 0.8991  | 0.9230    | 0.9092 |
> > > | 40%              | 0.9602  | 0.8917  | 0.9188    | 0.9015 |
> > > | 60%              | 0.9561  | 0.8893  | 0.9133    | 0.8928 |

---

### Official Review · Reviewer_uiLH · 2025-10-27

**Soundness:** 2
**Presentation:** 2
**Contribution:** 2
**Rating:** 2
**Confidence:** 4

**Summary:**

This paper proposes MemoPhishAgent, a memory-augmented multi-modal LLM agent framework for phishing URL detection. The system leverages five specialized tools orchestrated dynamically by an agent that incorporates an episodic memory for retrieving historical reasoning trajectories. MemoPhishAgent is evaluated on both public datasets and a new social-media-based phishing dataset, and the results show it outperforms several SOTA baselines in terms of recall and F1 score.

**Strengths:**

1. The proposed approach introduces an agent architecture enabling dynamic tool selection and multi-step reasoning, which is a promising direction for phishing detection.
2. The integration of episodic memory potentially enhances efficiency and adaptability by leveraging previous analysis experiences.
3. The inclusion of a real-world social media dataset makes the method more relevant for practical scenarios.
4. The ablation and sensitivity studies provide valuable insights into module contribution.

**Weaknesses:**

I believe the most significant weakness of this paper is the lack of transparency regarding the implementation details of the entire approach, as well as insufficient analysis of abnormal experimental observations.
1. It is unclear why the crawl content tool parses HTML into markdown before analysis. Since the goal is to analyze keywords, plain HTML could be sufficient. The manuscript does not provide experimental justification demonstrating that markdown processing is superior to directly using HTML.
2. The overall implementation process described in the paper is non-transparent. For example, important details such as the specific design and operation of each tool, the formats of inputs and outputs, the versions of the LLMs used, and the accuracy or reliability of individual tools are all missing or unclear. This lack of transparency makes it very difficult to assess the reliability and reproducibility of the proposed approach.
3. The authors do not publish the details of their own SocPhish dataset or provide access for reproducibility. Table 1 shows that MemoPhishAgent achieves much better results on the SocPhish dataset compared to baselines, but the advantage is far less on public datasets (TR-OP and DynaPD), and for some metrics, baselines even outperform the proposed method. This raises concerns about generalization and data representativeness.
4. The reasons why the Monolithic LLM architecture performs worse than the deterministic workflow agent are not well analyzed or explained. Additionally, there are cases where the Monolithic LLM matches or surpasses the proposed method, but the authors do not discuss these results.
5. It is not specified what engine or data source is used for intelligent search. If an external web search is applied, there is a risk that the search may directly retrieve the ground-truth answer (e.g., whether a URL is phishing), rather than relying solely on agent-based inference. This could affect the fairness and validity of the experimental evaluation. The paper does not discuss this risk or explain what steps were taken to avoid answer leakage during intelligent search.
6. The episodic memory module continuously accumulates past cases and reuses them in majority voting, but there is no mechanism to ensure the accuracy or correctness of previously stored decisions. If erroneous verdicts enter the memory, the majority-vote scheme might amplify these errors, potentially leading to false positives or negatives in future detections. The paper does not describe any strategy for memory cleansing, error correction, or aging out irrelevant experiences.

**Questions:**

1. The main innovation and contribution relative to previous agent-based or multi-modal phishing detectors is not sufficiently clarified. What exactly is improved over works such as PhishAgent or other agentic approaches?

---

> ### Author Response · Authors · 2025-11-24
> **Author Responses - Part 1**
>
> We thank the reviewer for their insightful suggestions and questions. We have responded to their individual points below.
>
> **The reviewer noted that the motivation for converting HTML to markdown before analysis is unclear. They questioned why plain HTML is not sufficient for keyword extraction and pointed out the lack of evidence to show that markdown improves analysis compared to using raw HTML.**
>
> We thank the reviewer for their insightful questions. First, we would like to clarify that our paper does not claim “Markdown is better than HTML” for phishing detection or LLM processing. The purpose of the crawl content tool is to (i) fetch the webpage, and (ii) expose the main textual content to the LLM in a compact, human-readable format so that the agent can reason about page semantics (e.g., login prompts, brand names) rather than just raw tags. Raw HTML contains substantial irrelevant content, e.g., tags, JavaScript, CSS, and navigation elements, which increases token usage and dilutes useful signals.
>
> Specifically, our system uses [Crawl4AI](https://github.com/unclecode/crawl4ai) as an off-the-shelf crawler, whose default behavior converts HTML to an LLM-friendly Markdown representation. We adopted this standard configuration because it is widely used in LLM-based RAG and agent systems [1], allowing us to focus on the agent’s reasoning rather than implementing custom parsing. Thus, Markdown is an implementation detail of the crawler, not a methodological assumption.
>
> To directly address the reviewer’s concern, we added a comparison experiment in which we vary only the crawl-content representation:
> - HTML: the tool returns the raw HTML of the webpage.
> - Cleaned text: the tool returns plain text extracted from HTML (tags removed).
> - Markdown (ours): the original setting using Crawl4AI’s Markdown output.
>
> We report phishing-detection metrics (accuracy, precision, recall, F1) and the average input token count of the crawl content tool on the SocPhish dataset. The results in Table 1 below show that (i) the detection performance is very similar across HTML and Markdown, and (ii) the Markdown and cleaned-text variants reduce the average number of tokens compared to raw HTML, but cleaned plain text degrades F1 and recall by a few points, suggesting that stripping all structure removes useful cues such as headings and hyperlinks. Overall, Markdown offers the best trade-off between performance and efficiency, and the small gaps across all three settings confirm that our framework is robust to the specific choice of crawl-content representation.
> Based on this, we conclude that our framework is robust to the choice of crawl content representation and that using Markdown via Crawl4AI is a practical engineering choice, not a core assumption of our method. We will clarify this point and add the analysis and all implementation details in the revised PDF.
>
> **Table 1. Comparison of HTML and Markdown for crawl content tool design.**
> | Format        |   ACC   |   F1    | Precision | Recall | Tokens |
> |---------------|---------|---------|-----------|--------|--------|
> | HTML          | 0.9642  | 0.9012  | 0.9270    | 0.9110 | 6590   |
> | Cleaned text  | 0.9589  | 0.8891  | 0.9123    | 0.8675 | 3312   |
> | Markdown      | 0.9657  | 0.9034  | 0.9257    | 0.9144 | 3873   |
>
> [1] Chen et al., MDEval: Evaluating and Enhancing Markdown Awareness in Large Language Models, WWW 2025.

---

> ### Author Response · Authors · 2025-11-24
> **Author Responses - Part 2**
>
> **The reviewer pointed out that the implementation details are insufficiently transparent. The reviewer asked for information, including tool design and functionality, input/output formats, LLM versions, and tool reliability.**
>
>
> We thank the reviewer for raising this concern. Section 3.3 describes the detailed design of our five tools, including their inputs, outputs, and running examples. Due to space limits, the full prompts for each tool, which detail their input/output formats, are provided in Appendix A.1.2. All tools are implemented as typed LangChain tools (using Pydantic schemas) and interact with the agent via JSON-serializable Python dictionaries. The backbone LLM used in our system is Claude 3.0.
>
> To directly address the reviewer’s question regarding tool design details, we will expand the Appendix in the revised PDF to include the full operational details of each tool, including input/output schemas and the reliability mechanisms we employ. To address the concern about transparency, we provide the complete agent implementation in [**this anonymous repo**](https://github.com/memophishagent/MemoPhishAgent.git).
>
> For reliability, we enforce typed inputs, explicit length caps to keep prompts LLM-safe, multiple URL variants (http/https), similarity thresholds for retrieval, and conservative fallbacks when crawling or JSON parsing fails, across all tools. These safeguards ensure that tools consistently return well-formed outputs rather than raising exceptions, allowing the agent to remain stable even when encountering malformed or adversarial webpages. To further evaluate tool reliability, we conduct two robustness tests under conditions common in phishing detection:
>
> - Stress test on malformed URLs.
> We curated 50 malformed URLs (invalid domains, unreachable hosts, unsupported schemes, random strings) and ran the agent on this set. The expected output is “Benign” with the reason “URL is invalid.” The agent achieved perfect accuracy with zero false positives. We also logged exceptions: none of the tools raised uncaught exceptions; all returned safe, structured fallback responses. This confirms that our safeguards successfully prevent tool-level failures on adversarial inputs.
>
> - Robustness to malformed LLM outputs.
> For tools that parse LLM-generated JSON (e.g., extract_targets_tool, judge_crawled_page, judge_image, check_screenshot), we simulate worst-case scenarios by injecting malformed or non-JSON outputs. For our five tools, on the dataset with 100 URLs, every time we randomly select one tool and inject corrupted LLM JSON output, with probability p, we try p=0.3 and p=0.5. As shown in Table 2 below, even when we intentionally inject malformed URLs or corrupt a significant fraction of intermediate LLM tool outputs (p = 0.3-0.5), the overall performance of MemoPhishAgent remains relatively unchanged, with accuracy and F1 decreasing by less than 1-2% and recall staying consistently high. The only noticeable effect is a modest increase in latency, caused by the agent’s built-in retry and fallback logic, which ensures safe recovery without raising exceptions. These results demonstrate that our system is resilient to both noisy inputs and faulty intermediate tool outputs, validating the reliability mechanisms built into our tool-calling framework. We will incorporate these clarifications and robustness results into the revised version.
>
> **Table 2. Reliability test of MemoPhishAgent tools on SocPhish.**
> | Condition                            |   ACC   |   F1    | Precision | Recall | Latency (s) | Exceptions |
> |--------------------------------------|---------|---------|-----------|--------|---------|------------|
> | Malformed URLs                       | 0.9635  | 0.8990  | 0.9235    | 0.9080 | 42.38   | 0          |
> | Malformed LLM outputs, p=0.3         | 0.9601  | 0.8932  | 0.9204    | 0.9031 | 43.78   | 0          |
> | Malformed LLM outputs, p=0.5         | 0.9550  | 0.8860  | 0.9151    | 0.8935 | 45.96   | 0          |
> | Original                             | 0.9657  | 0.9034  | 0.9257    | 0.9144 | 38.21   | 0          |

---

> ### Author Response · Authors · 2025-11-24
> **Author Responses - Part 3**
>
> **The reviewer noted that the SocPhish dataset is insufficiently described and unreleased, limiting reproducibility, and that the method performs better on SocPhish than on public datasets, where baselines sometimes outperform it.**
>
> We thank the reviewer for the comment. As stated in the paper, due to business constraints, we cannot release the full dataset details at this time. However, we will publish the dataset upon acceptance, pending the required internal approval. We outlined the key collection steps in lines 264–270: we designed phishing-related queries, performed large-scale scraping across five social media and forum platforms, and submitted the crawled URLs to a third-party verification service. We would also like to clarify that this situation is not unique to our work. [2] similarly does not release its full code or dataset due to commercial considerations. We are open to being contacted and are willing to facilitate dataset or code access under appropriate internal review and compliance steps.
>
> The performance advantage on SocPhish is due to two main factors. First, note that the SocPhish dataset is more challenging and realistic than the other two public datasets, as it contains live, in-the-wild URLs from social media with verified labels, reflecting true user exposure. Stronger performance on this dataset highlights the true effectiveness of our agent. Second, TR-OP and DynaPD are static and non-interactive, so we disabled the tool for extracting sublinks and used only four tools for those datasets. This naturally lowers performance and aligns with our ablation study showing that each tool contributes to accuracy.
>
> Regarding evaluation priorities, we explained in Section 4.1 that recall is the primary operational metric in our phishing-URL detection: all flagged URLs are forwarded to downstream third-party verification, so missing a true phishing site is far more costly than reviewing additional benign ones, considering the business loss. Across all three datasets, our agent achieves the highest recall, matching real deployment needs.
>
> Precision differences on the public datasets arise mainly from differences in dataset construction rather than limitations of our method. TR-OP and DynaPD are static and manually curated. Their pages are cleaner, more uniform, and often templated. SocPhish contains live, diverse, and obfuscated phishing attacks from real social media posts. In such realistic settings, the agent must act more cautiously when encountering ambiguous content, leading to slightly lower precision on cleaner benchmarks while still achieving the highest recall across all datasets.
>
> **The reviewer noted that the paper doesn’t explain why the monolithic baseline underperforms the deterministic workflow agent or address cases where the monolithic LLM outperforms the proposed method.**
>
> We thank the reviewer for the thoughtful questions. First, we would like to clarify that the two baselines exhibit complementary strengths rather than one being strictly better than the other. Monolithic LLM achieves better precision and latency than deterministic, while deterministic has better ACC, F1, and recall. The monolithic LLM has higher precision and lower latency because it tends to produce conservative “benign” predictions when uncertain and avoids tool calls entirely. However, aggregating all evidence (HTML+screenshot) into a single prompt limits its ability to reliably isolate subtle phishing cues, cross-check signals across modalities, or perform structured analysis. As a result, it frequently misses malicious URLs, which leads to its significantly lower recall. In contrast, the deterministic workflow agent achieves better ACC, F1, and recall by exhaustively executing all tools in a fixed, pre-defined order, ensuring that no modality is skipped and reducing the chance of missing a malicious sample. This behavior is consistent with Section 4.2: exhaustive tool usage increases sensitivity to phishing indicators, but at the cost of reduced precision and flexibility.
>
> As discussed in Section 4.2, the deterministic agent achieves higher recall by exhaustively executing all tools and following a fixed decision logic, which reduces the chance of overlooking malicious URLs. In contrast, the monolithic LLM aggregates all evidence into a single prompt and may miss key signals due to the long context.
>
> Our method outperforms both baselines by combining their strengths and mitigating their weaknesses. It performs multi-step reasoning with tool outputs (unlike the monolithic LLM) and adaptively selects or retries tools with episodic memory (unlike the deterministic agent). This adaptability yields the best overall performance, especially in recall, the most critical metric since missing a phishing URL is far costlier than reviewing extra benign ones.
>
> [2] Li et al., KnowPhish: Large Language Models Meet Multimodal Knowledge Graphs for Enhancing Reference-Based Phishing Detection, USENIX 2024.

---

> ### Author Response · Authors · 2025-11-24
> **Author Responses - Part 4**
>
> **The reviewer asked which engine powers the intelligent search and raised the concern that external search might leak ground-truth labels (e.g., whether a URL is phishing), compromising fairness. They noted that the paper does not discuss this risk or any safeguards against such leakage.**
>
> We appreciate the reviewer’s insightful questions. Our intelligent search is powered by Google Search via SerpAPI. As described in lines 182-186, the agent first proposes search queries; these queries are then executed through SerpAPI, which returns the formatted first-page search results.
>
> For phishing detection, there is not always a publicly available, 100% reliable ground-truth label for the URLs we evaluate; this is precisely why we rely on third-party takedown services during dataset construction. When performing an online search, no single external source directly provides a definitive “phishing” label for these newly crawled, realistic URLs. Therefore, external search cannot leak ground-truth answers through a single returned entry; at most, it provides indirect, noisy, and incomplete signals such as social-media discussions or security blog posts, which the agent must still interpret.
> Importantly, aggregating such weak external signals is exactly the purpose of the intelligent-search tool. When multiple search results consistently indicate suspicious activity, the agent can terminate early with high confidence. To ensure fair comparisons, all baselines receive equivalent access to the same search information: the deterministic agent includes the same search tool in its fixed pipeline, and the monolithic LLM baseline embeds the search results directly into its prompt. Thus, all methods benefit equally from any external signals, and no approach has privileged access to ground-truth labels.
>
> **The reviewer noted that the episodic memory continually stores past cases without verifying their correctness. The paper does not describe any mechanisms for memory validation, correction, or removal of outdated experiences.**
>
>
> We thank the reviewer for the insightful question. As noted in the paper, third-party verification requires 5-7 days, making real-time ground-truth labeling infeasible in an online phishing-detection setting. This is one motivation behind our majority-vote mechanism, which is designed to tolerate noise. The agent never relies on a single memory trace to make a final decision. Every classification is grounded in real-time evidence from the structured tool pipeline (crawling, visual analysis, intelligent search, etc.). Memory plays a supporting role by helping the agent recognize recurring patterns (e.g., repeated scam templates or impersonation motifs), and the strong empirical performance of MemoPhishAgent demonstrates that the memory system is effectively capturing useful signals.
>
> To directly address the reviewer’s concern, we designed a time-window pruning to gradually remove the outdated experiences of memory, under our current setup, where we will remove the least recently used 20%, 40% and 60% of the trajectories. Our preliminary results performance remains stable for forgetting strategies, demonstrating that our agent is robust to memory forgetting. We will leave the design of a more complex forgetting mechanism as our future work.
>
> **Table 1. Detection performance under different forgetting levels.**
> | Forgetting Level |   ACC   |   F1    | Precision | Recall |
> |------------------|---------|---------|-----------|--------|
> | No forgetting    | 0.9657  | 0.9034  | 0.9257    | 0.9144 |
> | 20%              | 0.9638  | 0.8991  | 0.9230    | 0.9092 |
> | 40%              | 0.9602  | 0.8917  | 0.9188    | 0.9015 |
> | 60%              | 0.9561  | 0.8893  | 0.9133    | 0.8928 |

---

> > ### Author Response · Authors · 2025-11-24
> > **Author Responses - Part 5**
> >
> > **The reviewer raised the concern that the paper did not clearly articulate its novelty over prior agent-based or multimodal phishing detectors. They questioned what concrete improvements have been made beyond systems like PhishAgent and other agentic approaches.**
> >
> > We thank the reviewer for this comment. To the best of our knowledge, PhishAgent is the closest agentic approach to our work. While both systems use multiple modalities, PhishAgent follows a static and predetermined tool sequence, where the order and use of tools are fixed by the pipeline design. In contrast, our MemoPhishAgent uses a ReAct-based adaptive tool selection mechanism, allowing the LLM to dynamically decide which tools to call, in what order, how many times, and whether to skip or repeat tools based on intermediate observations. This enables context-driven optimization and significantly reduces unnecessary tool calls, an ability not present in PhishAgent.
> >
> > Moreover, MemoPhishAgent incorporates episodic memory and in-context pattern retrieval, allowing the agent to leverage historical trajectories and emerging phishing templates during decision-making. PhishAgent does not include any memory component and does not reuse past experience to guide future predictions. Our method also supports multi-hop reasoning over tool outputs, whereas PhishAgent processes each tool output in a unidirectional, non-interactive manner. Thus, our method is structurally distinct and is behaviorally superior in adaptability, and operational grounding to the existing works.
> >
> > We will include all of the above clarifications and experiments in the revised version.

---

> ### Comment · Reviewer_uiLH · 2025-11-25
>
> Thanks for providing more details. However, I think the most relevant baseline is PhishAgent [1], as you have mentioned, and there is no comparison with PhishAgent. More importantly, even though PhishAgent doesn't release the code, they report the performance of PhishAgent on the TR-OP dataset, and it seems PhishAgent outperforms MemoPhishAgent on that dataset. Additionally, its time cost is far less than that of MemoPhishAgent. As a result, the effectiveness of MemoPhishAgent remains to be further verified.
>
> [1] Cao et al., PhishAgent: A Robust Multimodal Agent for Phishing Webpage Detection, AAAI'2025

---

> ### Author Response · Authors · 2025-11-25
>
> We sincerely appreciate the reviewer’s suggestion. As noted in the paper, PhishAgent is conceptually related, and we attempted to include it as a baseline. However, PhishAgent does not release code, checkpoints, or sufficient implementation details for reproduction, and our attempts to contact the authors received no response. As a result, we are unable to run PhishAgent on any of our other datasets, including SocPhish and DynaPD, which prevents a fair, apples-to-apples comparison under identical settings, tool availability, and evaluation protocol.
>
> Regarding the reviewer’s statement that “PhishAgent outperforms MemoPhishAgent on TR-OP”, we clarify that this comparison does not hold for several reasons:
> - **Different experimental setups.** PhishAgent evaluates 4,000 URLs (2,000 benign + 2,000 phishing) on TR-OP, whereas we evaluate 10,000 URLs (5,000 benign + 5,000 phishing). Directly comparing numbers across mismatched test sets is not meaningful. Moreover, MemoPhishAgent achieves higher recall, which is our primary evaluation priority for realistic phishing detection.
> - **Different model paradigms.** PhishAgent’s reported latency (“time cost”) reflects a single monolithic multimodal forward pass, not an agent architecture with multi-step tool calls and reasoning. Thus, the comparison does not evaluate the same capability or task setting.
> - **Different design objectives.** MemoPhishAgent is built for realistic online phishing detection with tool-based reasoning, multi-hop analysis, and episodic memory, capabilities not supported by PhishAgent’s static classifier architecture.
>
> To ensure fairness, we compared MemoPhishAgent extensively with publicly available, reproducible state-of-the-art multimodal phishing detectors (PhishLLM, PhishIntention, MLLM). Our method consistently achieves:
> - Highest recall on DynaPD datasets (Table 2 below).
> - Best F1 and ACC on SocPhish and TR-OP (Section 4 Table 1 in the paper).
> - Comparable latency to other pipelines.
>
> **Table 2 Detection performance comparison on the DynaPD dataset.**
> | Model            |   ACC   |   F1    | Precision | Recall | Latency (s) |
> |------------------|---------|---------|-----------|--------|---------|
> | MemoPhishAgent   | 0.8280  | 0.8448  | 0.7697    | 0.9360 | 38.21   |
> | PhishLLM         | 0.8581  | 0.8433  | 0.9262    | 0.7740 | 39.28   |
> | PhishIntention   | 0.6730  | 0.5350  | 0.9079    | 0.3740 | 28.50   |
> | MLLM|   0.7553 | 0.7449 | 0.7143 | 0.7781| 11.86|
>
>
> Given the lack of code and non-reproducibility of PhishAgent, we cannot include a controlled comparison, and relying on their reported numbers would be methodologically unsound.
> We will gladly add a direct empirical comparison when the PhishAgent implementation becomes publicly available, and we explicitly state this commitment in the revised PDF. For transparency, we have publicly released our own implementation, and we will add a clarification on this baseline discussion in the revised version. We hope this clarifies the issue and addresses the reviewer’s concern.

---

> > ### Comment · Reviewer_uiLH · 2025-11-27
> >
> > Thank you for the detailed response. I acknowledge the reproducibility issues with PhishAgent.
> >
> > However, I disagree that latency comparison is invalid due to 'different model paradigms.' While your internal mechanisms differ, **both methods solve the same task: phishing detection**. End-to-end latency for producing a detection result is what matters in practice, regardless of underlying architecture.

---

> > > ### Author Response · Authors · 2025-11-27
> > >
> > > We thank the reviewer for their thoughtful follow-up response and are glad that our earlier clarification helped address part of the concern. We agree that end-to-end latency is important in practical deployment. However, we would like to clarify two key points.
> > >
> > > **First, directly comparing the latency numbers reported in our paper and in PhishAgent is unfair and methodologically unsound.** Even when evaluated on the same dataset (e.g., TR-OP), the two latency values were obtained under different, non-comparable experimental setups, using evaluation environments. Without running both systems under the same hardware, backend services, and system constraints, directly comparing the reported numbers would yield misleading conclusions.
> > >
> > > **Second, our system solves a more challenging variant of phishing detection, one that PhishAgent is unable to handle.** While both systems output a “phishing vs. benign” label, the operational task formulation is fundamentally different. PhishAgent performs static snapshot classification on already-extracted HTML and screenshots. In contrast, MemoPhishAgent is designed for real-world, in-the-wild phishing detection, where URLs are:
> > > - live and interactive,
> > > - dynamically rendered with JavaScript,
> > > - capable of triggering additional network requests,
> > > - filled with child links that must be selectively followed and analyzed recursively,
> > > - evolving over time rather than fixed artifacts.
> > >
> > > This matches the real threat environment for brands and business companies, where phishing URLs are in-the-wild, dynamic, and interactive, not static screenshots.
> > > Therefore, “both systems solve the same task” does not reflect the practical, operational differences. MemoPhishAgent addresses a broader, harder, and more realistic detection problem that PhishAgent is unable to handle.
> > >
> > > Finally, we also provide concrete steps that can further reduce our latency in deployment:
> > > 1. Parallel processing. Our experiments process URLs sequentially due to LLM API rate limits, but the agent architecture supports full parallelism. With an adequate API quota, multiple worker agents can process URLs concurrently, reducing effective per-URL latency substantially.
> > > 2. Cached rendering and incremental crawling. Enabling caching for previously seen benign pages and for domain-level metadata can avoid repeated rendering. For our current crawler, enabling caching saves ~2 seconds per URL. Domain-level caching can avoid redundant evaluations within a configured time window.
> > >
> > > Taking these practical engineering optimizations together, MemoPhishAgent’s latency can be substantially reduced in real deployments. We will include these clarifications in the revised version.

---

### Official Review · Reviewer_RcbR · 2025-10-29

**Soundness:** 3
**Presentation:** 4
**Contribution:** 3
**Rating:** 8
**Confidence:** 3

**Summary:**

This paper introduces MemoPhishAgent, a memory-augmented multi-modal LLM agent framework for phishing URL detection. The system dynamically orchestrates five specialized tools (crawl content, check screenshot, check image, intelligent search, and extract targets) to gather evidence for phishing detection. The key innovation is an episodic memory system that captures past reasoning trajectories and supports three retrieval modes: majority-vote for high-confidence decisions, in-context exemplars for guided prompting, and full ReAct for novel threats. The authors evaluate their approach on three datasets including a newly collected SocPhish dataset from social media platforms, demonstrating superior performance over state-of-the-art baselines with 27% improvement in recall while maintaining manageable latency.

**Strengths:**

Originality: The work presents a creative combination of episodic memory with multi-modal agent reasoning for phishing detection. The three-tier memory retrieval strategy (no match, partial match, full match) is innovative and well-motivated. The problem formulation of using historical reasoning trajectories to improve detection is novel in the cybersecurity domain.

Quality: The experimental design is comprehensive with proper ablation studies demonstrating the necessity of each component. The evaluation across multiple datasets provides good coverage, and the tool usage analysis (Figure 3a) offers valuable insights into agent behavior. The statistical reporting with mean and standard deviation enhances credibility.

Clarity: The paper is well-written with clear motivation and methodology. Figure 1 effectively illustrates the system architecture, and the three-tier memory retrieval strategy is explained clearly. The experimental setup and evaluation metrics are appropriate for the task.
Significance: The work addresses a critical cybersecurity challenge with practical implications. The introduction of the SocPhish dataset provides value to the research community by reflecting real-world phishing threats. The 27% improvement in recall represents a substantial practical advancement that could reduce successful phishing attacks.

**Weaknesses:**

Scalability Problems: The biggest issue is that this system is too slow for real-world use. Taking 38 seconds per URL means it can't handle the millions of URLs that companies process daily. The memory system keeps growing as it processes more URLs, but the authors don't explain how this affects performance over time or how much storage it needs. There's no plan for removing outdated information, which could make the system slower and less accurate as it fills up with old data.

Security Vulnerabilities: The system showed a concerning 11% drop in accuracy when attackers used simple prompt injection attacks. This suggests that determined attackers could easily fool the system by crafting special inputs. Since the system relies on multiple tools working together, errors in one tool could cascade through the entire process. Even worse, attackers might be able to "poison" the memory by getting the system to remember their malicious examples as legitimate patterns.

**Questions:**

How does the episodic memory system's performance scale with increasing memory size, and what memory management strategies can maintain effectiveness at production scale while handling enterprise traffic of 100K-1M URLs per day?

---

> ### Author Response · Authors · 2025-11-24
> **Author Responses - Part 1**
>
> We thank the reviewer for their insightful suggestions and questions. We have responded to their individual points below.
>
> **The reviewer raised the concern that the latency is impractical for real-world settings. They suggested analysing the ever-growing memory in terms of runtime or storage cost, and strategy memory pruning to remove outdated information.**
>
> We thank the reviewer for raising the concern regarding the scalability. Regarding the 38s latency, we clarify that in our experiment, we process each URL one by one in a sequential manner, which was constrained by our LLM API rate limits. In practice, our agent is designed to support parallel execution: with sufficient API quota, multiple worker instances can process URLs concurrently, allowing throughput to scale linearly with available resources. In a production deployment with adequate parallelism, the effective per-URL latency would be significantly reduced.
>
> Regarding the concern about ever-growing memory, we first outline our design choices that naturally constrain memory size, then summarize the experiments we added to explore memory optimization. As described in Section 3, only URLs with maximum confidence (score = 5) are saved to episodic memory. Our analysis shows that 80% of judgments reach this threshold. This selective storage not only maintains memory quality by excluding low-confidence cases but also ensures storage efficiency. In addition, we store compact keyword embeddings as retrieval keys rather than full webpage content, which further reduces storage overhead.
>
> To provide concrete evidence of storage efficiency:
> - At our production scale of 5,000 URLs/day with 80% memory retention, the system stores about 4,000 entries per day, or 360,000 entries over a 90-days. Assuming an average of ~1 KB per entry (URL + compact metadata), this corresponds to about 360 MB of raw storage.
> - Even scaling to 100,000 daily URLs with the same 80% retention would yield ~7.2 million entries over 90 days, or about 7.2 GB of raw data at 1 KB per entry. Even after accounting for database and index overhead, this remains well within practical limits for modern infrastructure.
>
> To further address the reviewer’s concern about removing outdated information, we implemented a time-window pruning strategy to gradually discard stale experiences from memory on the SocPhish dataset. Specifically, each memory entry is assigned a usage counter that counts whether the most recent URL run has retrieved this entry. After every 50 processed URLs, we prune the least recently used 20%, 40%, and 60% of stored trajectories separately. Our results in Table 1 below show that performance remains stable across all forgetting strategies, indicating that the agent is relatively robust to memory pruning while benefiting from reduced storage. We will explore more sophisticated forgetting mechanisms as part of future work.
>
> **Table 1. Detection performance under different forgetting levels.**
> | Forgetting Level |   ACC   |   F1    | Precision | Recall |
> |------------------|---------|---------|-----------|--------|
> | No forgetting    | 0.9657  | 0.9034  | 0.9257    | 0.9144 |
> | 20%              | 0.9638  | 0.8991  | 0.9230    | 0.9092 |
> | 40%              | 0.9602  | 0.8917  | 0.9188    | 0.9015 |
> | 60%              | 0.9561  | 0.8893  | 0.9133    | 0.8928 |

---

> > ### Author Response · Authors · 2025-11-24
> > **Author Responses - Part 2**
> >
> > **The reviewer raised the concern about the drop in accuracy under prompt-injection attacks targeting the tool that checks the screenshot. They further noted that the attacker could poison the episodic memory by inserting malicious examples labeled as benign.**
> >
> > We appreciate the reviewer’s thoughtful concerns. We can mitigate the prompt injection attack through several well-established defenses. For example, we can perform an extra step of image-grounded reasoning to make sure that the check screenshot tool makes its judgment based on the visual content rather than irrelevant text. We can also perform an OCR scan on the screenshot before it is sent to the LLM. We will include a discussion of these defenses in the revised PDF and note that the multi-tool architecture inherently limits the impact of prompt injection against any single tool. And we will leave the exploration of more complex defenses as our future work.
> >
> > Regarding the memory poisoning attack, we would like to emphasize that our system design naturally provides strong robustness against the memory poisoning attack. First, only evaluations with maximum confidence (score = 5) enter episodic memory, establishing a quality threshold that makes low-confidence poisoning attempts ineffective. Second, the three-tier retrieval strategy employs majority-vote aggregation when multiple similar cases exist ($k \geq 5$), providing natural robustness against outliers; a single poisoned entry cannot override consensus from legitimate historical cases. Third, the similarity threshold ($\tau$) prevents loosely related malicious entries from influencing unrelated evaluations. Lastly, in order to make the memory poisoning effective, the attacker needs to manipulate multiple entries, which further degrades the stealthiness of the attack, making detection far more likely.
> >
> >
> > Regarding the reviewer's concern about tool cascading errors: our ReAct reasoning framework iteratively invokes tools and reasons over each returned evidence before deciding the next action. The agent does not blindly propagate tool outputs; instead, it generates intermediate rationales that assess whether the evidence is sufficient or requires verification. When a tool provides suspicious, incomplete, or contradictory information, the agent can invoke complementary tools to cross-validate the results. The final verdict requires the agent to synthesize accumulated evidence across multiple reasoning steps, preventing any single tool error from directly determining the outcome. As such, our iterative reasoning process provides meaningful protection against single-point failures. An attack that simultaneously targets multiple tools would require a significantly stronger threat model, as the adversary would need to manipulate several independent components at once. This substantially increases the difficulty and reduces the practicality of such an attack.
> >
> >
> > **The reviewer asked about the scalability of the memory system and how to maintain the effectiveness of the memory system for larger URL traffic.**
> >
> > As discussed in Part 1 above, only evaluations with maximum confidence (score = 5) are saved to episodic memory to exclude low-confidence cases and ensure storage efficiency. Secondly, we store only compact keyword embeddings as the key for retrieval rather than complete web content, which further minimizes the storage footprint.
> >
> > To provide concrete evidence of storage efficiency: at our production scale of 5,000 URLs daily with 80% memory retention, the system stores about 4,000 entries per day, or 360,000 entries over a 90-day window. Assuming an average of ~1 KB per entry (URL plus compact metadata), that’s roughly 360 MB of raw data. Even scaling to 100,000 daily URLs with the same 80% retention would yield ~7.2 million entries over 90 days, or about 7.2 GB of raw data at 1 KB per entry. Even after accounting for database and index overhead, this remains well within practical limits for modern infrastructure.
> >
> > To maintain the effectiveness of the memory system, several strategies can be explored. First, memory pruning or forgetting strategy, as shown in Table 1, is a promising approach for optimizing memory under higher traffic loads. Second, we can investigate memory summarization: instead of storing full tool-call sequences and raw outputs, the agent can invoke an additional LLM step to distill each trajectory into concise key takeaways, further improving memory efficiency without sacrificing utility.

---

> > > ### Comment · Reviewer_RcbR · 2025-11-27
> > >
> > > Thank you for your response — it addresses part of my concern. However, please make sure to clearly acknowledge these limitations in the paper and discuss possible avenues for improvement.

---

> > > > ### Author Response · Authors · 2025-11-28
> > > >
> > > > We thank the reviewer for the helpful suggestion. We will revise the paper to explicitly acknowledge these limitations, incorporate the discussion about the prompt-injection attack, memory pruning, and latency. We are also happy to address any further concerns the reviewer may have.

---

### Author Response · Authors · 2025-12-01

We thank all of the reviewers for the constructive and insightful feedback. Below, we summarize our responses during rebuttal:

## **New Experiments and Analysis**

We added all experiments suggested by reviewers. Below, we give a summary. All experiment setups and results are included in **Appendix A.1.3 (highlighted in blue)**.

**Detection performance under different forgetting levels.** We evaluated the agent under multiple memory-forgetting settings (no forgetting, and pruning the least-recent 20/40/60%). Across all conditions, accuracy, F1, and recall remain stable with only minor drops at high forgetting levels, demonstrating that the agent’s phishing detection remains robust even when substantial portions of memory are removed **(Reviewer RcbR, uiLH, QobJ, 39jv)**.

**Robustness to crawl-content representation.**
We compared HTML, cleaned text, and Markdown as crawl-content formats. Performance stays almost identical across all variants. Markdown and cleaned text reduce the token count. Markdown offers the best efficiency–performance trade-off, confirming that our method is robust to representation choices **(Reviewer uiLH)**.

**Robustness to malformed LLM outputs.**
We injected corrupted tool call outputs with probabilities 0.3 and 0.5. Accuracy and F1 decreased by <1–2%, recall remained high, and no exceptions occurred. Only latency increased modestly due to fallback and retry logic. This experiment demonstrates strong resilience to noisy or invalid intermediate LLM outputs **(Reviewer uiLH)**.

**Comparison with reference-based baselines.**
We further added PhishIntention [1] as a reference-based baseline on DynaPD. Our agent outperforms both PhishLLM and PhishIntention, confirming strong performance across both SOTA LLM-based and reference-based methods **(Reviewer uiLH, QobJ)**.

**Robustness to paraphrased tool prompts.**
We paraphrased each tool’s system prompt using GPT-5. Across all paraphrased variants, accuracy and F1 remain nearly identical to the original prompts, showing that the agent’s behavior is stable under semantically equivalent prompt changes **(Reviewer QobJ)**.

**Sensitivity to noisy memory entries.**
We constructed clean memory and injected controlled noise by flipping 25% and 50% of memory entries. Our detection performance remains relatively strong under moderate corruption, confirming that noisy memory cannot directly force misclassification and that the agent is robust to imperfect memory **(Reviewer QobJ)**.

**PR-AUC, ROC-AUC, and cost-sensitive recall curves.**
We plot the PR-AUC, ROC-AUC, and recall@k vs cost curves for our agent across all datasets, for the results reported in Table 1 in the paper. Results show that our agent achieves consistently strong ROC-AUC and PR-AUC scores, approaching 0.99 on SocPhish and remaining high on TR-OP and DynaPD. The recall@k vs cost curve further shows that our method preserves high recall across a wide range of decision thresholds **(Reviewer QobJ)**.

---

> ### Author Response · Authors · 2025-12-01
>
> ## **Clarifications Summary**
> **Reviewer RcbR**
>
> We clarified that the reported latency is based on sequential processing under API rate limits, while our agent is designed for parallel execution, which allows for deploying under a parallel setup and further reduces per-URL latency in practice.
>
> We clarified that memory growth is naturally constrained by our system design, including selective storage and compact embeddings.
>
> By providing concrete estimates, we demonstrated that even at large production scales, the total memory footprint remains modest (hundreds of MBs to a few GBs) and well within practical limits.
>
> **Reviewer uiLH**
>
> We clarified that our paper does not argue that Markdown is superior to HTML for phishing detection. Our experiments show that Markdown, cleaned text, and HTML yield similar detection performance, with Markdown offering lower token consumption.
>
> We demonstrated the robustness of our system through stress tests on malformed URLs and corrupted LLM outputs.
>
> We explained the performance differences between baselines and why our method outperforms both.
>
> We clarified that all five tools are fully specified in Section 3.3 and Appendix A.1.2, and that the intelligent-search tool does not provide ground-truth labels—only weak, noisy external signals that the agent must interpret. These signals are shared equally across all baselines, ensuring a fair comparison.
>
> We also provided the link to our complete anonymous code repository for full transparency.
>
> We clarified the key differences between PhishAgent and our agent. PhishAgent follows a fixed, static tool pipeline, whereas our method supports dynamic ordering, repetition, and skipping. We demonstrated that our agent incorporates episodic memory, multi-hop reasoning, and pattern retrieval, capabilities absent in PhishAgent. As a result, our method is structurally distinct and behaviorally more adaptable and grounded.
>
> **Reviewer QobJ**
>
> We clarified that our work fundamentally differs from prior approaches and that our distinctions from the baselines are formally defined in the paper, with all methods using identical tools and evaluated under fully controlled conditions across three datasets.
>
> We compared performance using PR-AUC, ROC-AUC, and cost-sensitive recall curves on all datasets, demonstrating the advantages of our agent over baseline methods.
>
> We clarified that all reported results are aggregated over five independent runs. We demonstrated our method’s robustness to paraphrased tool prompts and reduced variance in evaluation.
>
>
> **Reviewer 39jv**
>
> We clarified that the reported latency reflects a sequential experimental setup and provided concrete strategies for reducing per-URL latency in practice without modifying the core architecture.
>
> We provided a concrete long-term plan for managing the episodic memory.
>
> [1] Liu et al., Inferring Phishing Intention via Webpage Appearance and Dynamics: A Deep Vision Based Approach, USENIX 2022.

---

### Meta-Review · Area_Chair_QCsr · 2026-01-09

**Summary:**

This paper proposes MemoPhishAgent, a memory-augmented multi-modal LLM agent for phishing URL detection that dynamically orchestrates multiple tools and leverages episodic memory to reuse past reasoning trajectories. Reviewers agreed that the problem is important and that the system is thoughtfully engineered, with strong empirical gains in recall on realistic datasets. However, despite extensive rebuttal efforts and added experiments, reviewers raised persistent concerns regarding the limited novelty, evaluation fairness, and practical deployability of the approach. I recommend rejection.

**Reviewer Concerns:**

The primary concern across reviewers is that the core contribution is incremental relative to existing agent-based and multimodal phishing detection systems. While the authors argued that dynamic tool orchestration and episodic memory distinguish their approach from prior work (e.g., static pipelines), reviewers found that these distinctions were largely conceptual and insufficiently validated through controlled comparisons with the most relevant baselines.

Additional concerns remain regarding scalability and latency, as the reported per-URL inference time is substantially higher than competing methods, raising questions about real-world applicability. Reviewers also expressed reservations about evaluation validity, including reliance on an unreleased proprietary dataset, potential leakage or bias introduced by external search tools, and limited baseline coverage. Although the authors addressed many technical questions and added robustness analyses, these additions did not fully resolve concerns.

**Reviewer Scores:**

Reviewer scores were mixed but overall below the acceptance threshold. Importantly, the AC thinks the issues of reviewers were not fully resolved after the rebuttal.

---

### Decision · Program_Chairs · 2026-01-26

Reject